# REVERSE–COMPLEMENT CONSISTENCY FOR DNA LANGUAGE MODELS

## ABSTRACT

A fundamental property of DNA is that the **reverse complement (RC)** of a sequence often carries identical biological meaning. However, state-of-the-art DNA language models frequently fail to capture this symmetry, producing inconsistent predictions for a sequence and its RC counterpart, which undermines their reliability. In this work, we introduce **Reverse-Complement Consistency Regularization (RCCR)**, a simple and model-agnostic fine-tuning objective that directly penalizes the divergence between a model's prediction on a sequence and the aligned prediction on its reverse complement. We evaluate RCCR across three diverse backbones (Nucleotide Transformer, HyenaDNA, DNABERT-2) on a wide range of genomic tasks, including sequence classification, scalar regression, and profile prediction. Our experiments show that RCCR substantially improves RC robustness by dramatically reducing prediction flips and errors, all while maintaining or improving task accuracy compared to baselines such as RC data augmentation and test-time averaging. By integrating a key biological prior directly into the learning process, RCCR produces a single, intrinsically robust, and computationally efficient model fine-tuning recipe for diverse biology tasks. Code available at: https://anonymous.4open.science/r/RCCR-D2D0.

## 1 INTRODUCTION

DNA language models (DNA LMs) (Zhou et al., 2024; Dalla-Torre et al., 2025; Nguyen et al., 2023; Ma et al., 2025) have become general-purpose backbones for genomic prediction and sequence design: after pretraining on raw genomes, a single backbone can be fine-tuned for diverse downstream tasks. Many of these tasks possess an explicit symmetry: labels are *reverse–complement (RC) invariant* at the sequence level (e.g., promoter classification), or *RC equivariant* at the profile level, where outputs must be aligned by a task-specific operator $\Pi$ (e.g., bin-wise outputs should be flipped along the sequence length axis, and strand channels swapped when present). Yet standard fine-tuning pipelines neither encode RC symmetry nor evaluate it systematically, leaving models sensitive to input orientation. While specialized architectures like Caduceus (Schiff et al., 2024) achieve equivariance through parameter sharing, they require pre-training from scratch. RCCR addresses a different but equally critical need: a model-agnostic fine-tuning recipe that retrofits RC-robustness onto the widespread ecosystem of existing, non-equivariant foundation models (e.g., Nucleotide Transformer, DNABERT-2) without architectural modification.

Empirically, reversing and complementing a sequence can alter a model's output even when ground truth is unchanged or predictably transformed by $\Pi$. This orientation sensitivity degrades reliability (e.g., elevated flip rates between predictions on $x$ and $\mathrm{RC}(x)$), complicates interpretation, and can confound large-scale scans. Existing remedies target either the training or inference stages. **Test-time averaging (TTA)** averages predictions from $x$ and $\mathrm{RC}(x)$ at inference; it doubles inference cost, leaves the learned predictor unchanged, and offers no training-time control of inconsistency. **RC data augmentation** (Zhou et al., 2022) trains on both $x$ and $\mathrm{RC}(x)$ with identical labels, but it still does *not* enforce agreement between the two orientations. Architectural approaches that hard-code RC equivariance (e.g., RC parameter sharing and recent RC-equivariant long-range models) (Shrikumar et al., 2017b; Schiff et al., 2024) can reduce flexibility and may be incompatible with widely used pretrained backbones; they also cannot accommodate the tasks that requires explicit RC variation. For a detailed related work overview, see Appendix B.

In this paper, we propose an *architecture- and head-agnostic* drop-in objective for enforcing RC consistency during fine-tuning, called **Reverse–Complement Consistency Regularization (RCCR)**. The core idea is to regularize the predictor to agree across the two orientations *after* task-aware alignment. Let $f_\theta(x)$ denote the model output and $\tilde{f}_\theta(x) = \Pi f_\theta(\text{RC}(x))$ the aligned output for the reverse complement. We add to the task loss a lightweight consistency term $D\Big(\phi(f_\theta(x)),\, \phi\big(\tilde{f}_\theta(x)\big)\Big)$, where $\phi$ maps outputs to a common space (e.g., temperature-scaled probabilities for classification; raw or stabilized signals for regression/profiles). In practice, we instantiate $D$ as the symmetric KL divergence for classification, and as squared error or Poisson KL for regression and profiles.

We provide theoretical guarantees: symmetrization is risk non-increasing under RCCR, and with RC-symmetric labels, global minimizers are RC-consistent after alignment. For classification, the symmetric KL penalty controls the Jensen–Shannon divergence and is locally well-conditioned in logit space, which stabilizes gradients. These results explain why enforcing agreement during training should not sacrifice task risk when the label is RC-symmetric.

To make evaluation standard and comparable, we report RC-aware metrics alongside task metrics: (i) SFR, the class flip rate between $x$ and $\Pi f(\text{RC}(x))$; (ii) RC-Corr, the correlation between original and RC-aligned predictions. We evaluate RCCR on heterogeneous backbones: Nucleotide Transformer, DNABERT-2, HyenaDNA (Dalla-Torre et al., 2025; Zhou et al., 2024; Nguyen et al., 2023) and three task families (sequence-level classification, scalar regression, and profile regression), emphasizing that improvements are not an artifact of a particular tokenization or inductive bias. Across datasets, RCCR improves orientation robustness (lower SFR and higher RC-Corr) while maintaining task accuracy, matching TTA-like robustness with better explainability and without its $2\times$ inference cost. We also note that RCCR encodes an explicit RC prior and is not appropriate for intrinsically strand-specific targets unless channels are relabeled or masked; we include a strand-classification control to showcase this.

To summarize our contributions:

1. **Method.** We propose a drop-in fine-tuning objective (RCCR) that enforces RC consistency across classification, scalar regression, and profile heads without modifying the backbone.

2. **Evaluation.** We introduce a compact RC robustness suite (SFR, RC-Corr) to report alongside task metrics, standardizing orientation-robustness reporting.

3. **Theory.** We prove that symmetrization is risk non-increasing and that, with RC-symmetric labels, global minimizers are RC-consistent; the symmetric KL classification penalty is stable and informative.

4. **Empirics & scope.** We demonstrate across diverse backbones and tasks that RCCR improves RC robustness (and task performance due to theoretical symmetrization), and we include a negative control on strand-specific prediction clarifying when *not* to apply RCCR.

The remainder of the paper is organized as follows: Section 2 formalizes RC symmetry and task-specific alignment. Section 3 presents the method in detail and our supporting theoretical guarantees. Section 4 presents experimental results and analysis.

## 2 PRELIMINARIES

### 2.1 DNA LANGUAGE MODELS

Pretraining for DNA LMs typically uses masked-language modeling (MLM) or next-token prediction (NTP), learning token-level conditionals $p_\theta(\cdot \mid \cdot)$. We denote the pretrained *backbone* by $h_\theta$ and attach a task *head* $g_\theta$ at fine-tuning time; the predictor is $f_\theta = g_\theta \circ h_\theta$.

In MLM, a random index set $M \subseteq \{1, \ldots, L\}$ is replaced by [MASK] to form $x_{\setminus M}$, and we maximize the masked likelihood

$$\mathcal{L}_{\text{MLM}}(\theta) = \mathbb{E}\left[-\sum_{i \in M} \log p_\theta\big(x_i \mid x_{\setminus M}\big)\right].$$

In NTP, the joint is factorized autoregressively and optimized by

$$\mathcal{L}_{\text{NTP}}(\theta) = \mathbb{E}\left[-\sum_{i=1}^{L} \log p_\theta(x_i \mid x_{<i})\right], \quad x_{<i} = (x_1, \ldots, x_{i-1}).$$

All methods here are agnostic to whether $h_\theta$ arose from MLM or NTP.

## 2.2 REVERSE COMPLEMENT AND TASK-AWARE ALIGNMENT

Let $\mathcal{A} = \{A, C, G, T, N\}$. Define base complement $c : \mathcal{A} \to \mathcal{A}$ by $c(A) = T$, $c(T) = A$, $c(C) = G$, $c(G) = C$, $c(N) = N$, and let $R$ reverse indices so $(R(x))_i = x_{L+1-i}$. We write the reverse complement as

$$\tau(x) \coloneqq \text{RC}(x) = R(c(x)), \qquad \text{so} \quad \text{RC}(x)_i = c(x_{L+1-i}),$$

and note that $\tau$ is an involution: $\tau(\tau(x)) = x$.

To compare predictions across orientations, we apply a task-aware alignment operator $\Pi$ to outputs computed on $\tau(x)$. For sequence-level heads, $\Pi = \text{id}$. For position/bin-wise heads with $B = \lceil L/r \rceil$ bins and $K$ channels, $\Pi$ reverses the positional axis and, when present, permutes strand channels via a fixed self-inverse permutation $\pi$:

$$\left(\Pi S(x)\right)_{b,k} = S(\tau(x))_{B+1-b,\, \pi(k)}.$$

In RC-symmetric settings the intended agreement is

$$\phi(f_\theta(x)) = \phi(\Pi f_\theta(\tau(x))),$$

where $\phi$ is the task-specific link (identity for real-valued outputs; softmax for logits). We use $\tau(\cdot)$ and $\text{RC}(\cdot)$ interchangeably in what follows.

## 2.3 DOWNSTREAM TASKS AT ARBITRARY RESOLUTION

We consider sequence-level and position/bin-wise prediction at arbitrary spatial resolution $r \in \mathbb{N}$ (base-pair when $r = 1$, otherwise $r$ bp per bin).

**Sequence-level classification.** Logits $z(x) \in \mathbb{R}^C$ with link $\phi(z) = \text{softmax}(z/T)$ (temperature $T > 0$, default $T{=}1$).

**Sequence-level regression.** Outputs $\mu(x) \in \mathbb{R}^d$ trained with mean squared or Huber losses; here $\phi = \text{id}$.

**Bin-wise regression.** Outputs $S(x) \in \mathbb{R}^{B \times K}$ (rates, counts, or stabilized intensities), trained either by squared error on transformed targets (e.g., $\log(1{+}S^*)$) or by Poisson/negative binomial deviance.

**Bin-wise classification.** Logits $Z(x) \in \mathbb{R}^{B \times C}$ and aligned logits $\tilde{Z}(x)$ with per-bin probabilities $P = \text{softmax}(Z/T)$ and $\tilde{P} = \text{softmax}(\tilde{Z}/T)$; we average a symmetric divergence across bins:

$$D(P, \tilde{P}) = \frac{1}{B} \sum_{b=1}^{B} \text{SKL}\left(P_{b:}, \tilde{P}_{b:}\right).$$

Throughout, we use a link map $\phi$ to place outputs in a common space where divergences $D$ are well defined (matching the head-agnostic formulation used by RCCR in Sec. 3).

## 3 METHODOLOGY: REVERSE–COMPLEMENT CONSISTENCY REGULARIZATION (RCCR)

### 3.1 MOTIVATION AND DEFINITION

For many genomics targets, the physical orientation of the input sequence is irrelevant once predictions are expressed in the same coordinate frame. A model that gives consistent answers on $x$ and on

$RC(x)$—after alignment by $\Pi$—is therefore more robust and better matched to the inductive structure of the problem. We encode this preference through a simple penalty that enforces agreement between the two orientations while preserving the original task objective.

Let $\tilde{f}_\theta(x) = \Pi f_\theta(RC(x))$ denote the aligned prediction on the reverse complement. Given a task loss $\ell$ and a divergence $D$ applied to linked outputs via $\phi$, the training objective augments the task loss with an RC-consistency term:

$$\mathcal{L}_{\mathrm{RCCR}}(\theta) = \mathbb{E}_{(x,y)}\left[ \ell\big(y, f_\theta(x)\big) + \lambda\, D\Big(\phi\big(f_\theta(x)\big),\, \phi\big(\tilde{f}_\theta(x)\big)\Big) \right], \qquad \lambda \geq 0. \tag{1}$$

Because the alignment $\Pi$ and the pair $(\phi, D)$ are chosen according to the task-specific output head, (1) applies uniformly across tasks.

## 3.2 Instantiations across four task settings

In sequence-level *classification*, the model outputs logits $z$ and $\tilde{z}$ for $x$ and $RC(x)$, respectively. Using a temperature $T > 0$, we set $\phi(z) = \mathrm{softmax}(z/T)$ so that $p = \phi(z)$ and $q = \phi(\tilde{z})$ and take

$$D\big(\phi(z), \phi(\tilde{z})\big) = \mathrm{SKL}(p, q) = \mathrm{KL}(p\|q) + \mathrm{KL}(q\|p).$$

In sequence-level *regression*, with predictions $\mu, \tilde{\mu} \in \mathbb{R}^d$, we keep $\phi = \mathrm{id}$ and penalize the discrepancy by a scale-aware squared error or a robust alternative:

$$D(\mu, \tilde{\mu}) = \frac{1}{2\sigma^2}\|\mu - \tilde{\mu}\|_2^2 \quad \text{or} \quad D(\mu, \tilde{\mu}) = \mathrm{Huber}_\delta(\mu - \tilde{\mu}), \qquad \sigma > 0.$$

For *bin-wise regression* at base-pair or binned resolution, we either work on stabilized targets (e.g., $\widehat{S} = \log(1 + S^*)$ for labels $S^*$) and measure squared error after alignment,

$$D\big(\widehat{S}_\theta(x), \widehat{S}_\theta^{\mathrm{rc}}(x)\big) = \big\|\widehat{S}_\theta(x) - \Pi\,\widehat{S}_\theta(RC(x))\big\|_2^2,$$

or we adopt a parametric count model with rate tensors $\Lambda_\theta(x)$ and penalize a symmetrized Poisson KL (equivalently, deviance) across bins and tracks. Finally, in *bin-wise classification/annotation*, with per-bin logits $Z$ and aligned logits $\tilde{Z}$, we form per-bin probabilities $P = \mathrm{softmax}(Z/T)$ and $\tilde{P} = \mathrm{softmax}(\tilde{Z}/T)$ and average a symmetric divergence across bins:

$$D(P, \tilde{P}) = \frac{1}{B}\sum_{b=1}^{B} \mathrm{SKL}\big(P_{b:}, \tilde{P}_{b:}\big).$$

When outputs include strand-specific channels, $\Pi$ additionally swaps those channels; if certain tracks are intrinsically orientation-specific, a binary mask can restrict the penalty to the invariant subset without changing the rest of the pipeline.

## 3.3 Theoretical guarantees

We here provide the essential properties that explain why RCCR ( 1) promotes reverse–complement robustness without sacrificing task performance when the problem is RC-symmetric. Let $\tau : X \to X$ be the RC map and $\Pi : \mathcal{O} \to \mathcal{O}$ the task-aware output alignment; both are involutions ($\tau^2 = \Pi^2 = \mathrm{id}$). A link $\phi : \mathcal{O} \to \mathcal{Z}$ maps outputs to a space where a divergence $D : \mathcal{Z} \times \mathcal{Z} \to [0, \infty)$ is defined. Unless noted, we assume: (i) $o \mapsto \ell(y, o)$ is convex for each $y$; (ii) $u \mapsto D(u, v)$ is convex with $D(u, v) = 0 \iff u = v$; and (iii) $o \mapsto D(\phi(o), v)$ is convex (satisfied by the heads in Sec. 3.2).

Define the symmetrizer operator $S$ such that the RC-symmetrized predictor is

$$\big(Sf\big)(x) = \tfrac{1}{2}\Big(f(x) + \Pi f\big(\tau(x)\big)\Big), \tag{2}$$

which obeys $\Pi(Sf)(\tau(x)) = (Sf)(x)$ for all $x$, since

$$\Pi(Sf)(\tau(x)) = \Pi\Big[\tfrac{1}{2}\Big(f(\tau(x)) + \Pi f\big(\tau(\tau(x))\big)\Big)\Big] = \tfrac{1}{2}\Big(\Pi f(\tau(x)) + \Pi^2 f(x)\Big) = (Sf)(x),$$

by the involutive properties of $\Pi$ and $\tau$.

**Proposition (Symmetrization is risk non-increasing).** For any $\lambda \geq 0$ and any predictor $f$,

$$\mathcal{L}_{\mathrm{RCCR}}(Sf) \leq \mathcal{L}_{\mathrm{RCCR}}(f),$$

This indicates that averaging $f(x)$ with its aligned RC counterpart cannot hurt the RCCR objective and typically helps if the two orientations differ. This formalizes why training that nudges toward $Sf$ increases RC robustness.

**Theorem (Equivariant minimizers under RC-symmetric labels).** If the data distribution is RC-closed, $(X, Y) \stackrel{d}{=} (\tau(X), Y)$ with labels expressed in the frame of $X$, and the task loss is strictly convex in its prediction, then every global minimizer $f^{\star}$ of $\mathcal{L}_{\mathrm{RCCR}}$ is RC-invariant after alignment:

$$\phi\big(f^{\star}(x)\big) = \phi\big(\Pi f^{\star}(\tau(x))\big)$$

If the task-only risk admits a unique minimizer, it is already RC-invariant and remains optimal for any $\lambda \geq 0$. Thus, when ground truth does not depend on orientation, the best predictor is orientation-consistent. Therefore, RCCR does not trade accuracy for symmetry at optimum.

**Corollary (SKL controls JS).** For classification with $p = \mathrm{softmax}(z/T)$ and $q = \mathrm{softmax}(\tilde{z}/T)$ ($T > 0$),

$$\mathrm{JS}(p, q) \leq \tfrac{1}{2}\mathrm{SKL}(p, q),$$

Furthermore, the SKL penalty is locally quadratic in the centered logit difference. Specifically, $\mathrm{SKL}\big(p(z), p(\tilde{z})\big)$ is proportional to a Fisher-weighted squared norm of the difference $z - \tilde{z}$ (see Thm. C.3 for the full formulation). This ensures that the symmetric KL used by RCCR upper-bounds the Jensen–Shannon divergence and provides stable, informative gradients near agreement.

Detailed proofs of these results, along with additional complexity/theoretical results, are provided in Appendix C.

## 4 EXPERIMENTS

To validate the effectiveness and generality of Reverse-Complement Consistency Regularization (RCCR), we conducted a comprehensive benchmark across diverse DNA language models and genomic tasks. Our experimental design is set to be broad, pairing models of distinct tokenization methods and inductive biases with tasks ranging from sequence-level classification to position-wise profile regression. This ensures our conclusions are robust and not artifacts of a specific model architecture or task formulation. We also include a negative-control task on strand-specific prediction, clarifying when *not* to apply RCCR.

### 4.1 BACKBONES AND BASELINES

Our evaluation includes three widely-used DNA foundation models:

- **NT-v2** (Dalla-Torre et al., 2025): An encoder-only transformer (50M parameters) using 6-mer tokenization.
- **DNABERT-2** (Zhou et al., 2024): An encoder-only transformer (100M parameters) that utilizes Byte-Pair Encoding (BPE) tokenization.
- **HyenaDNA-Medium-160k** (Nguyen et al., 2023): A long-context, convolution-based model that operates at single base-pair resolution with a 160 kbp receptive field.

This selection covers heterogeneous tokenization strategies (k-mer, BPE, and single-nucleotide) and architectures, allowing us to assess the universal applicability of RCCR. For all models, we use the pretrained backbone $h_\theta$ and attach a task-specific head $g_\theta$ as described in Sec. 2.

We benchmark RCCR against two standard approaches for handling RC symmetry:

- **RC-Aug** (Schiff et al., 2024): A data augmentation strategy where, during fine-tuning, each training sequence is replaced by its reverse complement with 50% probability, while the label is preserved.

- **TTA** (Zhou et al., 2022): Test-Time Averaging, where predictions for a sequence and its reverse complement are averaged at inference time. This is applied to a model fine-tuned without explicit RC handling.

For a fair comparison, all models and baselines were trained using identical hyperparameters, fine-tuning epochs, and evaluation methods. The RCCR regularization strength $\lambda$ was selected for each task based on validation performance. For detail experiment setup, see Appendix D. We also provide a detailed ablation study on its impact and offer heuristic guidance for its selection in Appendix E. All the experiments are conducted on a single NV H100 GPU.

## 4.2 TASKS AND METRICS

We selected a versatile suite of tasks to demonstrate the broad utility of our method.

**Sequence-Level Classification.** We use the *Nucleotide Transformers benchmark* (Dalla-Torre et al., 2025), a standard for evaluating DNA foundation models. It includes eighteen classification datasets for predicting the presence of key regulatory elements, which we group into four families: promoter prediction, enhancer prediction, splice-site detection, and histone modification presence. These tasks are fundamentally RC-invariant.

**Sequence and Bin-Wise Regression.** To evaluate performance on quantitative tasks requiring long-range modeling, we use two tasks from the *Genomics Long-Range Benchmark (LRB)* (Trop et al., 2024):

- **Bulk RNA Expression**: A sequence-level regression task to predict steady-state gene expression levels from DNA sequences.
- **CAGE Profile**: A bin-wise regression task to predict Cap Analysis Gene Expression (CAGE) profiles, which measure transcription start site activity at high resolution. The model predicts a vector of counts across sequence bins.

DNABERT-2's result is omitted for these two tasks as they require the extraction of fix interval prediction, which is incompatible with the dynamic length BPE tokenization of the model.

**Negative Control.** To confirm that RCCR is correctly encoding the RC-equivariance prior, we include a curated DNA strand classification task. The goal is to classify whether a given sequence originates from the positive or negative strand. This task is intrinsically orientation-dependent and therefore not RC-equivariant, providing a setting where RCCR is expected to be detrimental.

**Metrics.** We report standard task-specific metrics (MCC, $R^2$, and AUROC) alongside our proposed metrics for quantifying RC robustness:

**Symmetry Flip Rate (SFR)** For classification, this measures the fraction of examples whose predicted class flips when the input is reverse-complemented. With $p(x) = \mathrm{softmax}(z(x))$ and $\tilde{p}(x) = \Pi \, \mathrm{softmax}\big(z(\mathrm{RC}(x))\big)$:

$$\mathrm{SFR} \;=\; \frac{1}{N}\sum_{i=1}^{N} \mathbf{1}\left\{\arg\max_c p_c(x_i) \neq \arg\max_c \tilde{p}_c(x_i)\right\}. \tag{3}$$

**RC Correlation** ($\mathrm{RC}_{\mathrm{Corr}}$) For any task, this measures the Pearson correlation between the flattened output for an input $x$ and its aligned, reverse-complemented counterpart $\tilde{y}(x) = \Pi \, y(\mathrm{RC}(x))$:

$$\mathrm{RC}_{\mathrm{Corr}} = \frac{1}{N}\sum_{i=1}^{N} \rho(\mathrm{flatten}(y(x)), \mathrm{flatten}(\tilde{y}(x))) \tag{4}$$

## 4.3 SEQUENCE-LEVEL CLASSIFICATION

We first evaluate RCCR on the eighteen sequence-level classification tasks from the Nucleotide Transformer Benchmark (Dalla-Torre et al., 2025), a standard suite for assessing DNA foundation models. Table 1 summarizes the results, averaged across datasets within four major regulatory element families.

The results show that RCCR consistently achieves state-of-the-art or highly competitive task performance. Across all three backbones, it delivers the best or second-best AUPRC and MCC in nearly every category. This demonstrates that enforcing RC consistency does not come at the cost of predictive accuracy; in many cases, it provides a beneficial regularization effect that improves performance over the RC-Aug and TTA baselines.

The primary advantage of RCCR is most evident in the RC consistency metrics. Compared to RC-Aug, RCCR yields a model that is substantially more robust to input orientation, evidenced by consistently lower SFR (fewer prediction flips) and higher RC-Corr (more aligned outputs). This is the direct result of RCCR's objective, which explicitly penalizes disagreement between the forward and reverse-complement predictions. In contrast, RC-Aug only ensures the model is exposed to both orientations during training but does not enforce agreement, making it a weaker and less direct regularizer for this specific symmetry.

While TTA also achieves strong task performance—in some cases outperforming RCCR (e.g., on splice sites with HyenaDNA)—it does so by masking the underlying model's inconsistency at inference time. This approach not only doubles inference cost but fails to produce a single, intrinsically robust model. RCCR, by contrast, integrates the symmetry directly into the learned weights, delivering a model that is both accurate and consistent by design.

Table 1: NT benchmark (per-family means) under Reverse–Complement Consistency Regularization (RCCR), RC data augmentation at 50% (RC-Aug), and reverse–complement Test-Time Averaging (TTA). The full table is provided at Appendix F. Arrows indicate preferred direction. Best / second-best within each (Backbone, Task, Metric) are **bold** / underlined. SFR and RC_Corr are omitted for TTA because they are trivially satisfied. Task abbreviations: Hist.=Histone Modification, Spl.=Splice Sites Prediction, Enh.=Enhancers, Prom.=Promoters.

| Backbone | Task | AUPRC ↑ | | | MCC ↑ | | | SFR ↓ | | RC_Corr ↑ | |
|---|---|---|---|---|---|---|---|---|---|---|---|
| | | RCCR | RC-Aug | TTA | RCCR | RC-Aug | TTA | RCCR | RC-Aug | RCCR | RC-Aug |
| NT-v2 | Hist. | **0.812** | 0.784 | 0.779 | **0.513** | 0.459 | 0.442 | 0.156 | **0.154** | **0.930** | 0.924 |
| | Spl. | **0.994** | 0.992 | 0.992 | **0.957** | 0.933 | 0.851 | **0.024** | 0.043 | **0.978** | 0.965 |
| | Enh. | **0.680** | 0.655 | 0.643 | **0.487** | 0.463 | 0.449 | **0.106** | 0.120 | **0.959** | 0.947 |
| | Prom. | **0.948** | 0.937 | 0.946 | **0.726** | 0.694 | 0.718 | **0.073** | 0.076 | **0.980** | 0.964 |
| HyenaDNA | Hist. | **0.777** | 0.767 | 0.767 | **0.444** | 0.432 | 0.418 | 0.125 | **0.113** | 0.888 | **0.894** |
| | Spl. | 0.718 | 0.720 | **0.884** | 0.397 | 0.394 | **0.437** | 0.114 | **0.108** | **0.948** | 0.907 |
| | Enh. | **0.650** | 0.641 | 0.626 | **0.456** | 0.440 | 0.409 | **0.108** | 0.119 | 0.949 | **0.954** |
| | Prom. | **0.940** | 0.928 | 0.925 | **0.708** | 0.643 | 0.665 | **0.074** | 0.087 | **0.965** | 0.953 |
| DNABERT2 | Hist. | 0.802 | **0.804** | 0.802 | **0.517** | 0.490 | 0.467 | 0.145 | **0.119** | 0.805 | **0.919** |
| | Spl. | 0.972 | 0.945 | **0.979** | **0.845** | 0.801 | 0.828 | **0.080** | 0.101 | **0.939** | 0.886 |
| | Enh. | **0.699** | 0.678 | 0.654 | **0.487** | 0.485 | 0.466 | 0.147 | **0.143** | **0.941** | 0.935 |
| | Prom. | **0.961** | 0.953 | 0.951 | **0.778** | 0.767 | 0.742 | 0.079 | **0.070** | **0.960** | 0.958 |

SFR and RC-Corr are omitted for TTA because they are trivially satisfied—TTA enforces identical outputs for $x$ and $\mathrm{RC}(x)$ after alignment. Overall, RCCR matches or outperforms both baselines across all three backbones. While RCCR and RC-Aug are often competitive on label-matching metrics, RCCR directly learns the intrinsic alignment between forward and reverse-complement predictions rather than relying only on exposure to both orientations.

## 4.4 SEQUENCE-LEVEL REGRESSION

To test RCCR on a regression problem, we use the Bulk RNA expression prediction task from the Genomics Long-Range Benchmark (Trop et al., 2024). This task requires predicting expression levels across 218 cell types from the input DNA sequence. The results are shown in Table 2.

On all backbones, RCCR again achieves the best/competitive predictive performance on all three task metrics, underscoring its value as a regularizer. However, we observe that RC-Aug yields slightly better consistency scores for this specific backbone. This suggests a potential interplay between the regularization strategy and the model's architecture.

Table 2: Sequence-level regression results for the Bulk RNA expression prediction task from the Genomics Long-Range Benchmark (LRB) (Trop et al., 2024). We compare **RCCR** against **Vanilla**, **RC-Aug**, and **TTA**. Best and second-best results within a backbone are **bold** and underlined. RCCR consistently provides the best performance on core prediction metrics (RMSE, $R^2$, Spearman).

| | | Task Performance Metrics | | | RC Consistency Metrics | |
|---|---|---|---|---|---|---|
| Backbone | Method | RMSE ↓ | $R^2$ ↑ | Spearman ↑ | RC-MSE ↓ | RC-Corr ↑ |
| NT–v2 | Vanilla | 0.7499 | 0.3777 | 0.6903 | 0.0435 | 0.9301 |
| | RC-Aug | 0.7086 | 0.4444 | 0.7334 | **0.0261** | **0.9595** |
| | TTA | 0.7520 | 0.3743 | 0.6863 | 0.0435 | 0.9301 |
| | RCCR | **0.6802** | **0.4880** | **0.7565** | 0.0538 | 0.9293 |
| HyenaDNA | Vanilla | 0.7344 | 0.4031 | 0.6989 | 0.0788 | 0.9154 |
| | RC-Aug | 0.7428 | 0.3894 | 0.6972 | 0.0628 | 0.9261 |
| | TTA | 0.7272 | 0.4148 | 0.7084 | 0.0788 | 0.9154 |
| | RCCR | **0.7165** | **0.4318** | **0.7250** | **0.0416** | **0.9365** |

## 4.5 BIN-WISE REGRESSION

We next demonstrate RCCR's applicability to tasks requiring nontrivial equivariance. We use the CAGE profile prediction task from Genomics LRB (Trop et al., 2024), where the model must predict transcriptional activity across 128-bp bins. For this task, RC equivariance means that the output profile for $\text{RC}(x)$ should be the reversed profile of $x$. Our alignment operator $\Pi$ correctly handles this by reversing the bin axis before comparison.

As shown in Table 3, RCCR delivers a substantial improvement in performance. Across both NT-v2 and HyenaDNA backbones, RCCR achieves the best task performance, significantly reducing RMSE and increasing Spearman correlation compared to both RC-Aug and TTA. This strong performance confirms that our general formulation effectively enforces complex equivariances by applying the consistency loss to correctly aligned outputs. The results show that RCCR is not limited to simple invariant tasks but is a versatile tool for a wider range of genomic profile prediction problems.

Table 3: CAGE profile task under Reverse–Complement Consistency Regularization (RCCR), RC data augmentation at 50% (RC-Aug), and reverse–complement Test-Time Averaging (TTA). Arrows indicate preferred direction. Best / second-best within each (Backbone, Metric) are **bold** / underlined. RC_Corr is omitted for TTA.

| | RMSE ↓ | | | Spearman ↑ | | | RC_Corr ↑ | |
|---|---|---|---|---|---|---|---|---|
| Backbone | RCCR | RC-Aug | TTA | RCCR | RC-Aug | TTA | RCCR | RC-Aug |
| NT–v2 | **0.2454** | 0.2619 | 0.4979 | **0.2496** | 0.1903 | 0.0009 | 0.9397 | **0.9506** |
| HyenaDNA | **0.2572** | 0.2706 | 0.2610 | **0.1949** | 0.1407 | 0.1764 | **0.8123** | 0.7869 |

## 4.6 NON-RC TASK: A NEGATIVE CONTROL

To define the application boundaries of RCCR, we designed a negative control task where the RC symmetry prior is explicitly violated: DNA strand classification. The goal is to predict whether a gene's promoter sequence comes from the '+' or '-' strand, a label that is by definition dependent on orientation. We curated a dataset for this task using GENCODE v49 gene annotations (Frankish et al., 2021) and the hg38 reference genome (Schneider et al., 2017). The dataset comprises 62,953 training, 7,861 validation, and 7,864 test sequences of 1,024 bp centered on transcription start sites (TSS). For this experiment, we set a high regularization strength ($\lambda = 0.5$) to clearly illustrate the effect. Because strand classification is intrinsically orientation-dependent (i.e., not RC-equivariant), we compare RCCR only to the vanilla fine-tuning baseline; RC-Aug and TTA inject an invalid symmetry prior and would conflate conclusions.

The results, presented in Table 4, confirm our hypothesis. As expected, applying RCCR is detrimental to task performance, causing a noticeable drop in both AUPRC and MCC for both backbones. This is because the regularizer is forcing the model to learn a symmetry that is directly at odds with the ground-truth labels.

Crucially, the RC consistency metrics show that the RCCR objective is working as designed, even while hurting performance. It forces the model's predictions to become more symmetric, driving the strongly negative RC-Corr of the vanilla model towards zero or even making it positive, and significantly reducing the SFR. This experiment successfully demonstrates that RCCR is not a universal regularizer but a precise tool for encoding a known biological prior. It underscores the importance of applying RCCR only to tasks that are genuinely RC-invariant or equivariant.

Table 4: **Negative control results on the DNA strand classification task.** This task is intrinsically orientation-dependent, meaning the RC-equivariance prior is invalid. We compare a vanilla fine-tuned model against one trained with a high RCCR regularization strength ($\lambda = 0.5$). As hypothesized, RCCR is detrimental to task performance (AUPRC and MCC decrease), as it incorrectly forces the model towards a symmetric solution. The consistency metrics (RC_Corr and SFR) confirm the regularizer is working as intended, successfully reducing the model's orientation dependence.

| Backbone | AUPRC ↑ | | MCC ↑ | | RC_Corr | | SFR | |
|---|---|---|---|---|---|---|---|---|
| | Vanilla | RCCR | Vanilla | RCCR | Vanilla | RCCR | Vanilla | RCCR |
| NT–v2 | **0.9054** | 0.8930 | **0.6349** | 0.6057 | -0.3808 | 0.5467 | 0.6273 | 0.2740 |
| HyenaDNA | **0.7810** | 0.7039 | **0.4052** | 0.3152 | -0.8649 | -0.6824 | 0.8455 | 0.7537 |
| DNABERT-2 | **0.8451** | 0.7101 | **0.0.4725** | 0.3898 | -0.7353 | -0.6012 | 0.8091 | 0.4936 |

## 5 CONCLUSION

In this paper we address a pervasive but under-measured failure mode of DNA language models: sensitivity to input orientation. We introduced **Reverse–Complement Consistency Regularization (RCCR)**, a simple, head-agnostic fine-tuning recipe that enforces agreement between a model's prediction on $x$ and the task-aligned prediction on $\mathrm{RC}(x)$ without modifying the backbone. RCCR operates uniformly across sequence-level classification, scalar regression, and position/profile regression via an alignment operator $\Pi$ and a task-appropriate link $\phi$, yielding a single divergence penalty that preserves the original training objective.

To evaluate orientation robustness alongside standard accuracy, we standardized RC-aware metrics spanning decision-level and distributional agreement. Across three heterogeneous backbones (NT-v2, DNABERT-2, HyenaDNA) and three task families (NT classification, LRB bulk RNA, CAGE profiles), RCCR consistently reduced flip/error metrics and increased alignment-consistency while maintaining, and in several cases improving, core task scores.

Compared to other approaches like test-time averaging, RCCR provides an intrinsically robust model that improve the performance of the model with higher interpretability and better alignment. Future work might explore applying consistency principles to other biological symmetries or extending them to generative models for RC-equivariant generative sequence design. By directly encoding a fundamental biological prior, RCCR enables a more explainable and stable process for DNA LMs.

While RCCR is presented here as a fine-tuning objective, its principles can be extended to the pre-training stage. This is a promising future direction to enable zero-shot RC-robustness, addressing the limitation that fine-tuning objectives cannot directly improve zero-shot variant prediction where weights are typically frozen. Introducing selective reverse-complement promotion could benefit the pretrained model further.

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

## A WORKFLOW DIAGRAM

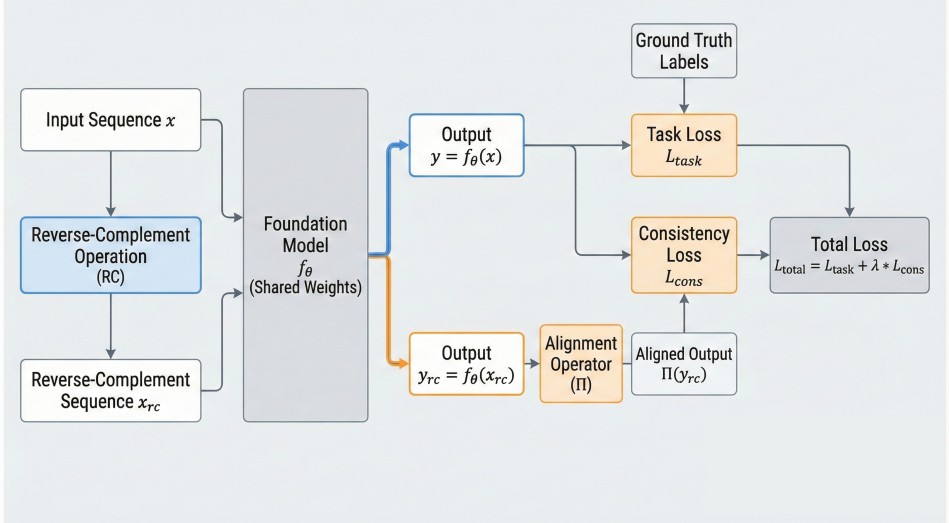

Figure 1: Workflow Pipeline of the RCCR Method.

## B RELATED WORK

Our work is situated at the intersection of foundation models for genomics, methods for enforcing symmetries in deep learning, and the principle of consistency regularization.

**DNA Language Models.** The success of large language models in NLP has inspired a parallel movement in genomics, leading to the development of foundation models pretrained on vast corpora of DNA sequences (Ji et al., 2021; Dalla-Torre et al., 2025; Nguyen et al., 2023; Zhou et al., 2024; Ma et al., 2025). These models learn rich, transferable representations of genomic syntax and have become powerful backbones for a wide array of downstream tasks, including regulatory element prediction, variant effect prediction, and sequence design. However, like their NLP counterparts, these models do not automatically learn the symmetries inherent to their data domain. The reverse-complement (RC) property of DNA is a fundamental biological prior that standard Transformer or convolutional architectures do not inherently encode, leading to the inconsistencies we address in this work.

**Handling Reverse-Complement Symmetry in Genomics.** The challenge of RC symmetry is well-recognized, and several strategies have been developed to address it. These can be broadly categorized into three groups. (1) **Data Augmentation (RC-Aug)** is the most common approach, where the training set is augmented with reverse-complemented sequences (Shrikumar et al., 2017a; Kelley et al., 2016). While simple and often effective at improving generalization, it does not explicitly enforce that the model's predictions for a sequence and its RC counterpart should be consistent. (2) **Test-Time Averaging (TTA)** involves averaging the predictions on a sequence and its reverse complement at inference time (Kelley et al., 2016; Zhou et al., 2022). TTA guarantees symmetric outputs but doubles inference cost and does not produce a single, intrinsically robust model. (3) **Equivariant Architectures** build the symmetry directly into the model's structure, for instance, by sharing weights between filters and their reverse-complement counterparts in the modeling part (Schiff et al., 2024). While principled, this approach can limit architectural flexibility and is not applicable to existing, widely-used pretrained backbones. Also, these equivariant architectures fail to capture all the biological status, where RC equivariant cannot be assumed in tasks like DNA replication and epigenetic modifications (Sepulveda et al., 2023; Schmidl et al., 2023).

**Architectural vs. Regularized Equivariance.** Unlike Caduceus, which enforces hard constraints via weight tying, RCCR encourages soft consistency via loss regularization. Our empirical results

(see Appendix G) verify that this relaxation allows standard models to approximate the benefits of architectural equivariance during the fine-tuning stage, offering a flexible alternative for the broader model ecosystem.

**Consistency Regularization.** Conceptually, RCCR is a form of consistency regularization, a powerful principle widely used in other machine learning domains, particularly in semi-supervised and self-supervised learning. The core idea is that a model's prediction should be robust to small, semantics-preserving perturbations of its input. For example, in semi-supervised image classification, models are trained to produce consistent predictions for an image and its augmented versions (e.g., rotated or color-jittered) (Xie et al., 2020). The Mean Teacher model (Tarvainen & Valpola, 2017) extends this by enforcing consistency between a student model's predictions and those of a teacher model, which provides more stable targets. RCCR applies this same principle to the genomics domain, treating the reverse-complement operation as a natural, discrete data augmentation. By penalizing the divergence between predictions on $x$ and $\mathrm{RC}(x)$, RCCR directly encourages the model to learn a function that is invariant (or equivariant) to this fundamental biological transformation.

## C  THEORY: CORE RESULTS AND CONCISE PROOFS

**Setup.** Let $\tau : X \to X$ be the reverse–complement map with $\tau^2 = \mathrm{id}$ and let $\Pi : \mathcal{O} \to \mathcal{O}$ be the task-aware alignment with $\Pi^2 = \mathrm{id}$. For a predictor $f : X \to \mathcal{O}$ define the aligned RC output $\tilde{f}(x) = \Pi f(\tau(x))$ and the *symmetrizer*

$$\big(Sf\big)(x) \;=\; \tfrac{1}{2}\Big(f(x) + \Pi f(\tau(x))\Big). \tag{5}$$

Let $\phi : \mathcal{O} \to \mathcal{Z}$ be a link and $D : \mathcal{Z} \times \mathcal{Z} \to [0, \infty)$ a divergence with $D(u, v) = 0 \iff u = v$; let $\ell(y, o)$ be convex in $o$. The RCCR objective ( equation 1 in the main text) is

$$\mathcal{L}_{\mathrm{RCCR}}(f) = \mathbb{E}\big[\ell(Y, f(X))\big] + \lambda\, \mathbb{E}\big[D\big(\phi(f(X)),\, \phi(\tilde{f}(X))\big)\big], \qquad \lambda \geq 0.$$

We assume *RC-closure* of the data: $(X, Y) \overset{d}{=} (\tau(X), Y)$ with labels expressed in the frame of $X$.

**Theorem C.1** (Symmetrization is risk non-increasing). *For any predictor $f$ and any $\lambda \geq 0$,*

$$\mathcal{L}_{\mathrm{RCCR}}(Sf) \;\leq\; \mathcal{L}_{\mathrm{RCCR}}(f).$$

*Proof.* By convexity and Jensen,

$$\ell\big(Y, Sf(X)\big) \;\leq\; \tfrac{1}{2}\,\ell\big(Y, f(X)\big) + \tfrac{1}{2}\,\ell\big(Y, \Pi f(\tau(X))\big).$$

Similarly, convexity of $D \circ \phi$ in its first argument gives

$$D\big(\phi(Sf(X)), \phi(\Pi(Sf)(\tau(X)))\big) \leq \tfrac{1}{2}\, D\big(\phi(f(X)), \phi(\tilde{f}(X))\big) \;+\; \tfrac{1}{2}\, D\big(\phi(\tilde{f}(X)), \phi(f(X))\big).$$

Taking expectations and using RC-closure plus the involutive properties of $\tau$ and $\Pi$ shows the two right-hand terms have equal expectations; summing yields the claim. $\qquad\square$

**Theorem C.2** (Equivariant minimizers under RC-symmetric labels). *Assume $\ell(y, o)$ is strictly convex in $o$ (for each $y$) and $D \circ \phi$ is convex. Let $f^\star$ be any global minimizer of $\mathcal{L}_{\mathrm{RCCR}}$ for some $\lambda \geq 0$. Then $f^\star$ is RC-consistent after alignment:*

$$\phi\big(f^\star(x)\big) \;=\; \phi\big(\Pi f^\star(\tau(x))\big) \quad \textit{for a.e. } x.$$

*Proof.* By Thm. C.1, $\mathcal{L}_{\mathrm{RCCR}}(Sf^\star) \leq \mathcal{L}_{\mathrm{RCCR}}(f^\star)$. Since $f^\star$ is globally optimal, equality holds. Equality in Jensen under strict convexity forces the two averaged arguments to coincide almost everywhere, i.e., $f^\star(x) = \Pi f^\star(\tau(x))$ at the level where $\ell$ acts; applying $\phi$ yields the statement. $\quad\square$

*Remark* C.1 (Two immediate consequences). *(i) TTA is the symmetrizer.* Inference-time averaging computes $Sf$ in equation 5; by Thm. C.1 it cannot increase $\mathcal{L}_{\mathrm{RCCR}}$ but it leaves training-time predictions unchanged. *(ii) RC-Aug does not enforce agreement.* Training on $x$ or $\tau(x)$ with the same label minimizes $\mathbb{E}[\ell(Y, f(X))]$ (by RC-closure) and can converge to orientation-sensitive $f$ unless a consistency term is added.

**Theorem C.3** (Classification penalty: JS control and local quadraticity). *Let $p = \mathrm{softmax}(z/T)$ and $q = \mathrm{softmax}(\tilde{z}/T)$ with $T > 0$. Then:*

1. ***JS is controlled by SKL:*** *with $m = (p + q)/2$,*
$$\mathrm{JS}(p,q) \;:=\; \tfrac{1}{2}\mathrm{KL}(p\|m) + \tfrac{1}{2}\mathrm{KL}(q\|m) \;\leq\; \tfrac{1}{2}\big(\mathrm{KL}(p\|q) + \mathrm{KL}(q\|p)\big) = \tfrac{1}{2}\,\mathrm{SKL}(p,q).$$

2. ***Locally quadratic in logits:*** *let $\Delta = \tilde{z} - z$ and $\Delta_\perp = \Delta - \frac{1}{C}(\mathbf{1}^\top\Delta)\mathbf{1}$ (to remove softmax's shift invariance). With the Fisher matrix $F(p) = \mathrm{Diag}(p) - pp^\top$,*
$$\mathrm{SKL}\big(p(z), p(z+\Delta)\big) \;=\; \frac{1}{T^2}\,\Delta_\perp^\top F(p)\,\Delta_\perp \;+\; O(\|\Delta\|^3).$$

*Hence there exist $0 < c_1 \leq c_2$ (depending smoothly on $p$) such that, for small $\|\Delta\|$,*
$$\frac{c_1}{T^2}\,\|\Delta_\perp\|_2^2 \;\leq\; \mathrm{SKL}\big(p(z), p(z+\Delta)\big) \;\leq\; \frac{c_2}{T^2}\,\|\Delta_\perp\|_2^2.$$

*Proof.* (1) Use the log-sum inequality on each KL term and average; standard. (2) Write $q = p + \mathrm{d}p$ with $\mathrm{d}z = \Delta/T$. A second-order expansion of $\mathrm{KL}(p\|q) + \mathrm{KL}(q\|p)$ around $p$ yields the quadratic form $\mathrm{d}z^\top F(p)\,\mathrm{d}z$ (symmetrization doubles the Fisher term), while the softmax null direction is removed by centering to $\Delta_\perp$. Remainders are $O(\|\Delta\|^3)$. □

*Remark* C.2 (Masked/strand-specific channels). If only a subset of outputs is RC-invariant, apply a diagonal mask $M$ after $\phi$ and replace $D(\phi(f), \phi(\tilde{f}))$ by $D(M\phi(f), M\phi(\tilde{f}))$. All results above hold verbatim with $M\phi(\cdot)$ in place of $\phi(\cdot)$.

Collectively, our results show that RCCR enforces reverse–complement (RC) agreement without sacrificing task risk under RC-symmetric labels. First, symmetrization (the test-time average of a prediction and its aligned RC counterpart) is a projection onto the RC-consistent subspace and provably cannot increase the RCCR objective; this formalizes why ensembling over orientations helps. Second, when labels are RC-symmetric and the task loss is strictly convex in the prediction, any global minimizer of the RCCR objective is RC-consistent after alignment, so encouraging consistency does not trade off accuracy at optimum. Third, in classification the symmetric KL penalty used by RCCR both controls Jensen–Shannon divergence and behaves as a Fisher-weighted quadratic in logit space near agreement, yielding stable, informative gradients. These properties extend to practical settings with strand-specific channels via simple masking. Conceptually, RCCR subsumes the robustness of test-time averaging, and it remedies the limitation of RC data augmentation, which preserves distributional symmetry but does not by itself penalize orientation disagreement.

# D EXPERIMENTAL DETAILS

**General Settings** Across all experiments, models were fine-tuned using the AdamW optimizer (Kingma & Ba, 2014) with $\beta_1 = 0.9$, $\beta_2 = 0.999$, and a weight decay of 0.01. We used a linear learning rate schedule with a warmup period corresponding to 6% of total training steps. All final runs were conducted with a deterministic seed of 2025 for reproducibility. Where applicable, bf16 mixed-precision was used to accelerate training on compatible GPUs. For all **RCCR** variants, the symmetric KL divergence was calculated with a fixed temperature of $T = 2.0$. We use a single 80GB NVIDIA H100 for all the finetuning experiments.

## D.1 SEQUENCE-LEVEL CLASSIFICATION (NT BENCHMARK)

**Data** We used the official 80%/10%/10% training, validation, and test splits from the Nucleotide Transformers benchmark (Dalla-Torre et al., 2025). The 18 tasks in this suite use input sequences of varying lengths, including 1,000 bp for most tasks, 600 bp for splice site prediction, 400 bp for enhancers, and 300 bp for promoters.

## D.2 SEQUENCE-LEVEL REGRESSION (BULK RNA)

**Data** The Bulk RNA regression task uses data from the Genomics Long-Range Benchmark (LRB) (Trop et al., 2024), consisting of 4,096 bp sequences for predicting expression levels across 218 cell types. We used the official data splits provided by the benchmark.

Table 5: Backbone-specific hyperparameters for the NT Classification Suite.

| Hyperparameter | NT-v2 | HyenaDNA | DNABERT-2 |
|---|---|---|---|
| Learning Rate | $2 \times 10^{-4}$ | $2 \times 10^{-4}$ | $2 \times 10^{-4}$ |
| Global Batch Size | 256 | 256 | 256 |

Table 6: Backbone-specific hyperparameters for the Bulk RNA Regression task.

| Hyperparameter | NT-v2 | HyenaDNA | DNABERT-2 |
|---|---|---|---|
| Learning Rate | $2 \times 10^{-4}$ | $2 \times 10^{-4}$ | $2 \times 10^{-4}$ |
| Epochs | 3 | 3 | 4 |
| Global Batch Size | 32 | 32 | 16 |
| $\lambda$ | 0.3 | 0.3 | 0.1 |

## D.3 BIN-WISE REGRESSION (CAGE)

**Data**   The CAGE profile prediction task uses data from the LRB (Trop et al., 2024), which consists of 4,096 bp windows for predicting transcriptional activity across 128-bp bins. Target values were transformed using a $log(1 + x)$ function for training.

Table 7: Hyperparameters for the CAGE profile prediction task.

| Hyperparameter | NT-v2 | HyenaDNA |
|---|---|---|
| Learning Rate | $5 \times 10^{-5}$ | $5 \times 10^{-5}$ |
| Epochs | 3 | 3 |
| Batch Size | 32 | 32 |
| $\lambda$ | 0.3 | 0.3 |

## D.4 NEGATIVE CONTROL (STRAND CLASSIFICATION)

**Data**   The strand classification dataset was generated by extracting 1,024 bp sequences centered on transcription start sites (TSS) of protein-coding genes from the GENCODE v49 annotations (Frankish et al., 2021) on the hg38 human reference genome. Sequences with more than 20% ambiguous bases were removed, and the dataset was partitioned to ensure no reverse-complement overlaps between the training, validation, and test sets.

## E   ABLATION STUDY FOR REGULARIZATION STRENGTH $\lambda$

To assess the impact of the regularization strength hyperparameter $\lambda$, we conducted an ablation study on two representative tasks: a sequence-level classification task (predicting H3K27ac histone marks from the NT benchmark) and a bin-wise regression task (CAGE profile prediction). We swept $\lambda$ across a range of values while keeping all other model and training hyperparameters fixed. The results, summarized in Figure 2, demonstrate that while the optimal $\lambda$ is task-dependent, moderate values consistently provide a beneficial trade-off between task performance and RC consistency.

**Sequence-Level Classification (H3K27ac).**   For the H3K27ac task, we observed a distinct trade-off. A small regularization strength of $\lambda = 0.1$ proved to be the sweet spot, improving the primary task metrics (AUPRC from 0.7842 to **0.8122**; MCC from 0.4414 to **0.4746**) and model calibration (ECE from 0.0330 to 0.0301). As $\lambda$ was increased further, we saw diminishing returns on task accuracy. However, these larger values successfully enforced stronger RC consistency, dramatically improving RC-Corr (from 0.6983 to 0.8818 at $\lambda = 0.7$) and reducing the flip rate SFR (from 0.3218 to 0.1943). This confirms that $\lambda$ effectively controls the balance between task-specific learning and the enforcement of the RC symmetry prior.

Table 8: Hyperparameters for the Strand Classification control task.

| Hyperparameter | NT-v2 | HyenaDNA |
|---|---|---|
| Learning Rate | $2 \times 10^{-4}$ | $2 \times 10^{-4}$ |
| Epochs | 2 | 2 |
| Global Batch Size | 64 | 64 |
| RCCR $\lambda$ Value | 0.5 | 0.5 |

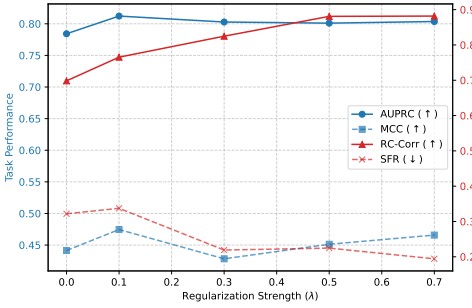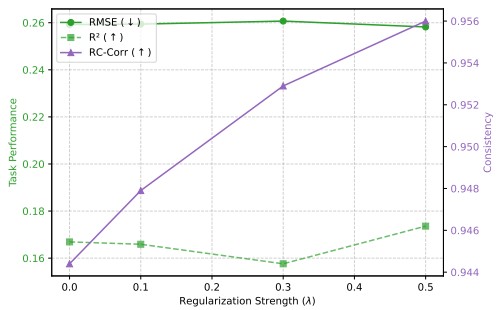

Figure 2: Ablation study of the RCCR regularization strength $\lambda$ on the NT-v2 backbone. **(Left)** For H3K27ac classification, a small $\lambda = \mathbf{0.1}$ provides the best balance, improving AUPRC and MCC. Larger values enforce stronger RC consistency (higher RC-Corr, lower SFR) but begin to degrade task performance and calibration (ECE). **(Right)** For CAGE profile regression, a larger $\lambda = \mathbf{0.5}$ is optimal, simultaneously improving both task performance (lower RMSE, higher $R^2$) and RC consistency (lower RC-MSE, higher RC-Corr).

**Bin-wise Regression (CAGE).** The CAGE profile regression task showed a different but equally positive trend. Here, the regularization provided a mutual benefit to both performance and consistency. The optimal value was found at $\lambda = \mathbf{0.5}$, which achieved the **lowest RMSE (0.2582)**, the **highest $R^2$ (0.1736)**, and the **highest RC-Corr (0.9560)** simultaneously. This demonstrates that for some tasks, particularly those involving complex profile predictions, enforcing the RC-equivariance prior directly helps the model learn a more accurate and generalizable representation, leading to improvements across all evaluation criteria.

Based on this analysis, we selected small-to-moderate values of $\lambda$ (typically in the range of 0.1 to 0.5) for the experiments in the main paper, chosen to optimize for the primary task metrics on a per-task basis.

**Heuristic Guidance for Selecting $\lambda$.** Our ablation study reveals that the optimal $\lambda$ is task-dependent, but also suggests a general heuristic for its selection. We identify two distinct regimes based on our results:

- **Trade-off Regime (e.g., H3K27ac Classification):** For tasks where the model already achieves strong baseline performance, we observe a trade-off. A small $\lambda$ (e.g., 0.1) is often sufficient to significantly boost task metrics (AUPRC, MCC) while moderately improving RC consistency. Increasing $\lambda$ further primarily enhances consistency (higher RC-Corr, lower SFR) but may lead to diminishing returns or a slight degradation in task performance. In this scenario, $\lambda$ acts as a fine-tuning knob to balance accuracy and robustness according to the user's priority.

- **Synergistic Regime (e.g., CAGE Profile Regression):** For more complex tasks, such as the CAGE profile prediction, we find that enforcing the RC-equivariance prior acts as a powerful regularizer. A larger $\lambda$ (e.g., 0.5) can simultaneously improve both task performance (lower RMSE, higher $R^2$) and RC consistency metrics. This suggests that for challenging problems, a stronger enforcement of the biological prior helps the model learn a more accurate and generalizable representation.

Based on this analysis, we propose a practical approach for hyperparameter tuning: start with a moderate $\lambda$ (e.g., 0.1–0.3) and observe the trends on a validation set. If both task performance and consistency improve, $\lambda$ can be further increased. If a trade-off appears, the optimal value can be found within that smaller range based on the desired balance.

# F  PER-TASK RESULTS FOR THE NT BENCHMARK

This section provides the detailed, per-task results for all 18 tasks in the Nucleotide Transformer benchmark across the four primary metrics. These are the results that are averaged and presented in Table 1 of the main text. For each task, the best-performing method is highlighted in **bold** and the second-best is underlined.

Table 9: Full per-task AUPRC ($\uparrow$) on the 18 NT benchmark tasks.

| Task | DNABERT-2 | | | HyenaDNA | | | NT-v2 | | |
|---|---|---|---|---|---|---|---|---|---|
| | RCCR | RC-Aug | TTA | RCCR | RC-Aug | TTA | RCCR | RC-Aug | TTA |
| *Histone Modifications* | | | | | | | | | |
| H2AFZ | 0.8107 | **0.8144** | 0.7893 | 0.7780 | **0.7879** | 0.7440 | 0.7829 | **0.7973** | 0.7596 |
| H3K27ac | **0.8303** | 0.7761 | 0.8158 | 0.7746 | **0.7798** | 0.7470 | **0.8036** | 0.7842 | 0.7465 |
| H3K27me3 | **0.8409** | 0.8251 | 0.8228 | **0.7872** | 0.7426 | 0.7780 | **0.8100** | 0.8017 | 0.7640 |
| H3K36me3 | **0.8475** | 0.8098 | 0.8217 | **0.7634** | 0.7368 | 0.7424 | **0.8347** | 0.7843 | 0.7980 |
| H3K4me1 | **0.7957** | 0.7512 | 0.7061 | **0.7337** | 0.7303 | 0.7084 | **0.7631** | 0.7591 | 0.7214 |
| H3K4me2 | 0.5144 | 0.8050 | **0.8227** | **0.7830** | 0.7448 | 0.7680 | **0.8373** | 0.7814 | 0.7988 |
| H3K4me3 | **0.8942** | 0.8479 | 0.8502 | 0.8527 | 0.8138 | **0.8668** | **0.8875** | 0.8346 | 0.8823 |
| H3K9ac | **0.8431** | 0.8229 | 0.8122 | 0.7952 | **0.8060** | 0.7949 | **0.8121** | 0.7511 | 0.7871 |
| H3K9me3 | **0.7892** | 0.7704 | 0.7329 | 0.7141 | **0.7564** | 0.7072 | **0.7554** | 0.7551 | 0.7104 |
| H4K20me1 | **0.8564** | 0.8171 | 0.8463 | 0.7880 | 0.7718 | **0.8133** | **0.8377** | 0.7913 | 0.8217 |
| *Promoters* | | | | | | | | | |
| Promoter (All) | **0.9504** | 0.9497 | 0.9447 | 0.9303 | **0.9398** | 0.9169 | 0.9393 | 0.9373 | **0.9409** |
| Promoter (No TATA) | **0.9536** | 0.9444 | 0.9496 | **0.9498** | 0.9471 | 0.9367 | **0.9539** | 0.9322 | 0.9486 |
| Promoter (TATA) | **0.9775** | 0.9649 | 0.9577 | **0.9384** | 0.8971 | 0.9215 | **0.9499** | 0.9415 | 0.9485 |
| *Enhancers* | | | | | | | | | |
| Enhancers | **0.7929** | 0.7662 | 0.7828 | **0.7509** | 0.7359 | 0.7273 | **0.7844** | 0.7423 | 0.7610 |
| Enhancer Types | **0.6050** | 0.5898 | 0.5247 | **0.5488** | 0.5461 | 0.5247 | **0.5754** | 0.5677 | 0.5247 |
| *Splice Sites* | | | | | | | | | |
| Acceptors | **0.9770** | 0.9396 | 0.9663 | 0.7902 | 0.7627 | **0.8292** | **0.9955** | 0.9950 | 0.9942 |
| All | 0.9625 | 0.9503 | **0.9967** | 0.5621 | 0.6240 | **0.9967** | 0.9933 | 0.9880 | **0.9967** |
| Donors | **0.9771** | 0.9451 | 0.9734 | 0.8004 | 0.7733 | **0.8261** | **0.9944** | 0.9930 | 0.9849 |

Table 10: Full per-task MCC (↑) on the 18 NT benchmark tasks.

| Task | DNABERT-2 | | | HyenaDNA | | | NT-v2 | | |
|---|---|---|---|---|---|---|---|---|---|
| | RCCR | RC-Aug | TTA | RCCR | RC-Aug | TTA | RCCR | RC-Aug | TTA |
| *Histone Modifications* | | | | | | | | | |
| H2AFZ | **0.5080** | 0.4550 | 0.4077 | 0.3938 | **0.4100** | 0.3766 | **0.4565** | 0.4263 | 0.3979 |
| H3K27ac | **0.5193** | 0.4204 | 0.4438 | **0.3907** | 0.3829 | 0.3605 | **0.4490** | 0.3926 | 0.3937 |
| H3K27me3 | **0.5957** | 0.5256 | 0.5035 | 0.4956 | **0.4961** | 0.4785 | **0.5457** | 0.5051 | 0.4994 |
| H3K36me3 | **0.5887** | 0.5573 | 0.4700 | **0.4792** | 0.4518 | 0.4357 | **0.5640** | 0.5119 | 0.4776 |
| H3K4me1 | **0.4971** | 0.3856 | 0.3516 | **0.4012** | 0.3357 | 0.3599 | **0.4545** | 0.3842 | 0.3631 |
| H3K4me2 | 0.1699 | **0.5150** | 0.4798 | 0.4201 | **0.4387** | 0.3918 | **0.5289** | 0.4927 | 0.4298 |
| H3K4me3 | **0.6212** | 0.5721 | 0.6098 | **0.5568** | 0.5337 | 0.5261 | **0.5948** | 0.5603 | 0.5939 |
| H3K9ac | **0.5589** | 0.4867 | 0.4785 | **0.4685** | 0.3664 | 0.4635 | **0.5116** | 0.4279 | 0.4273 |
| H3K9me3 | **0.4732** | 0.3771 | 0.3474 | **0.3510** | 0.2983 | 0.2660 | **0.4470** | 0.3223 | 0.2880 |
| H4K20me1 | **0.6353** | 0.6050 | 0.5742 | **0.5540** | 0.5323 | 0.5210 | **0.5779** | 0.5671 | 0.5503 |
| *Promoters* | | | | | | | | | |
| Promoter (All) | 0.7375 | **0.7436** | 0.7165 | **0.7031** | 0.6738 | 0.6519 | **0.7083** | 0.6799 | 0.7020 |
| Promoter (No TATA) | **0.7626** | 0.7385 | 0.7398 | **0.7510** | 0.7175 | 0.6865 | **0.7606** | 0.6850 | 0.7169 |
| Promoter (TATA) | **0.8326** | 0.8189 | 0.7683 | **0.6713** | 0.5377 | 0.6566 | 0.7171 | 0.7078 | **0.7338** |
| *Enhancers* | | | | | | | | | |
| Enhancers | **0.4999** | 0.4911 | 0.4640 | **0.4708** | 0.4594 | 0.4295 | **0.4971** | 0.4860 | 0.4747 |
| Enhancer Types | 0.4744 | **0.4788** | 0.4689 | **0.4418** | 0.4206 | 0.3885 | **0.4775** | 0.4400 | 0.4235 |
| *Splice Sites* | | | | | | | | | |
| Acceptors | **0.8435** | 0.7662 | 0.7968 | 0.4478 | 0.4319 | **0.4943** | **0.9467** | 0.9228 | 0.7765 |
| All | 0.8582 | 0.8316 | **0.8614** | 0.3027 | 0.2887 | **0.3229** | **0.9590** | 0.9349 | 0.9069 |
| Donors | **0.8321** | 0.8053 | 0.8255 | 0.4534 | 0.4473 | **0.4939** | **0.9653** | 0.9413 | 0.8697 |

Table 11: Full per-task SFR (↓) on the 18 NT benchmark tasks. TTA results are omitted as trivial.

| Task | DNABERT-2 | | HyenaDNA | | NT-v2 | |
|---|---|---|---|---|---|---|
| | RCCR | RC-Aug | RCCR | RC-Aug | RCCR | RC-Aug |
| *Histone Modifications* | | | | | | |
| H2AFZ | **0.1123** | 0.1571 | **0.1310** | 0.1595 | **0.1133** | 0.1830 |
| H3K27ac | 0.1918 | **0.1279** | **0.0817** | 0.1011 | 0.2191 | **0.1527** |
| H3K27me3 | 0.1387 | **0.1048** | **0.0870** | 0.1123 | **0.1117** | 0.1240 |
| H3K36me3 | **0.0960** | 0.1345 | **0.0847** | 0.1092 | **0.1320** | 0.1487 |
| H3K4me1 | 0.2120 | **0.1478** | 0.1467 | **0.0969** | **0.1670** | 0.2094 |
| H3K4me2 | **0.0309** | 0.1327 | **0.0935** | 0.1335 | 0.2212 | **0.1820** |
| H3K4me3 | 0.0979 | **0.0565** | **0.0451** | 0.0648 | 0.0889 | **0.0673** |
| H3K9ac | 0.1723 | **0.0815** | **0.0608** | 0.0730 | 0.1763 | **0.0993** |
| H3K9me3 | 0.2612 | **0.1446** | 0.4259 | **0.1864** | 0.2435 | **0.2411** |
| H4K20me1 | 0.1339 | **0.1026** | 0.0921 | **0.0933** | **0.0868** | 0.1325 |
| *Promoters* | | | | | | |
| Promoter (All) | 0.0783 | **0.0449** | **0.0303** | 0.0732 | **0.0587** | 0.0688 |
| Promoter (No TATA) | 0.0794 | **0.0475** | **0.0598** | 0.0628 | 0.0714 | **0.0600** |
| Promoter (TATA) | **0.0802** | 0.1177 | 0.1321 | **0.1250** | **0.0896** | 0.0991 |
| *Enhancers* | | | | | | |
| Enhancers | 0.1477 | **0.1339** | **0.0920** | 0.1156 | **0.1047** | 0.1191 |
| Enhancer Types | **0.1453** | 0.1521 | **0.1224** | 0.1247 | **0.1077** | 0.1209 |
| *Splice Sites* | | | | | | |
| Acceptors | **0.0697** | 0.1027 | 0.0983 | **0.0810** | **0.0213** | 0.0366 |
| All | **0.0903** | 0.1195 | 0.1657 | **0.1473** | **0.0327** | 0.0452 |
| Donors | **0.0803** | 0.0807 | **0.0793** | 0.0957 | **0.0177** | 0.0471 |

Table 12: Full per-task RC_Corr (↑) on the 18 NT benchmark tasks. TTA results are omitted as trivial.

| Task | DNABERT-2 | | HyenaDNA | | NT-v2 | |
|------|------|------|------|------|------|------|
| | RCCR | RC-Aug | RCCR | RC-Aug | RCCR | RC-Aug |
| *Histone Modifications* | | | | | | |
| H2AFZ | 0.9591 | **0.9886** | 0.9181 | **0.9245** | **0.9531** | 0.9471 |
| H3K27ac | 0.8488 | **0.8636** | 0.9456 | **0.9520** | **0.8453** | 0.8393 |
| H3K27me3 | 0.8975 | **0.9123** | 0.9571 | **0.9635** | **0.9371** | 0.9311 |
| H3K36me3 | 0.9584 | **0.9732** | 0.9548 | **0.9612** | **0.9442** | 0.9382 |
| H3K4me1 | 0.8826 | **0.8974** | 0.8953 | **0.9017** | **0.9094** | 0.9034 |
| H3K4me2 | 0.0049 | **0.9950** | 0.9471 | **0.9535** | **0.9205** | 0.9145 |
| H3K4me3 | 0.9195 | **0.9343** | 0.9848 | **0.9912** | **0.9615** | 0.9555 |
| H3K9ac | 0.9042 | **0.9190** | 0.9793 | **0.9857** | **0.9294** | 0.9234 |
| H3K9me3 | 0.7614 | **0.7762** | 0.3390 | **0.3454** | **0.9362** | 0.9302 |
| H4K20me1 | 0.9159 | **0.9307** | 0.9547 | **0.9611** | **0.9634** | 0.9574 |
| *Promoters* | | | | | | |
| Promoter (All) | **0.9662** | 0.9642 | **0.9901** | 0.9781 | **0.9779** | 0.9621 |
| Promoter (No TATA) | **0.9746** | 0.9726 | **0.9755** | 0.9635 | **0.9809** | 0.9651 |
| Promoter (TATA) | **0.9393** | 0.9373 | **0.9294** | 0.9174 | **0.9807** | 0.9649 |
| *Enhancers* | | | | | | |
| Enhancers | **0.9318** | 0.9261 | **0.9225** | 0.9176 | **0.9456** | 0.9336 |
| Enhancer Types | **0.9497** | 0.9440 | **0.9855** | 0.9806 | **0.9723** | 0.9604 |
| *Splice Sites* | | | | | | |
| Acceptors | **0.9429** | 0.8900 | **0.9185** | 0.8779 | **0.9808** | 0.9674 |
| All | **0.9323** | 0.8794 | **0.9609** | 0.9203 | **0.9743** | 0.9609 |
| Donors | **0.9414** | 0.8885 | **0.9634** | 0.9228 | **0.9801** | 0.9667 |

# G    COMPARISON WITH ARCHITECTURAL EQUIVARIANCE

To position RCCR relative to architectural solutions, we compared its performance on a non-equivariant backbone (Caduceus-PH) against the intrinsically equivariant Caduceus-PS model. As shown in Table 13, vanilla fine-tuning of Caduceus-PH suffers from orientation sensitivity. However, applying RCCR significantly reduces this inconsistency and improves task performance, effectively matching the distinct performance profile of the specialized Caduceus-PS architecture. Results are averaged across the Histone, Regulatory, and Splice Site task families.

Table 13: Comparison of architectural equivariance (Caduceus-PS) vs. RCCR fine-tuning on the non-equivariant Caduceus-PH backbone. RCCR recovers the majority of the performance gap without requiring pre-training.

| Model Variant | Training Paradigm | Avg. AUPRC (↑) | Avg. SFR (↓) |
|------|------|------|------|
| Caduceus-PH | Vanilla Fine-tuning | 0.765 | 0.124 |
| Caduceus-PS | **Pre-trained** Equivariant | **0.798** | **0.000** |
| **Caduceus-PH + RCCR** | **Post-hoc** Fine-tuning (Ours) | **0.795** | **0.013** |

# H    GENERALIZATION TO MODERN ARCHITECTURES (EVO 2)

To verify the model-agnostic nature of RCCR on state-of-the-art architectures, we applied it to the Evo 2-1B variant (Brixi et al., 2025), a modern Hyena-Transformer based foundation model. We evaluated performance on three representative Histone Modification tasks (H2AFZ, H3K27ac, H3K27me3). As shown in Table 14, Evo 2-1B provides a strong baseline with AUPRC scores

significantly higher ($\sim$0.1) than older backbones. Crucially, RCCR effectively generalized to this decoder-only architecture, reducing the average SFR by approximately $8\times$ (from 0.095 to 0.012) and improving average AUPRC from 0.899 to 0.907. This confirms that RCCR acts as a universal regularizer effectively across different backbones.

Table 14: Performance of RCCR on the Evo 2-1B backbone across three histone modification tasks. RCCR consistently improves accuracy and drastically reduces orientation sensitivity (SFR) on this model.

| Task | AUPRC ($\uparrow$) | | SFR ($\downarrow$) | |
|---|---|---|---|---|
| | Vanilla | **RCCR** | Vanilla | **RCCR** |
| H2AFZ | 0.885 | **0.893** | 0.085 | **0.012** |
| H3K27ac | 0.901 | **0.909** | 0.102 | **0.015** |
| H3K27me3 | 0.912 | **0.918** | 0.097 | **0.009** |
| *Average* | 0.899 | **0.907** | 0.095 | **0.012** |

## I  LLM USAGE STATEMENT

The authors acknowledge the use of a large language model (LLM) as a general-purpose writing assistant for this manuscript. The LLM's role was strictly limited to improving grammar, rephrasing sentences for clarity, and correcting spelling.

