# OpenReview forum: "Reverse-Complement Consistency for DNA Language Models"
_ICLR.cc/2026/Conference — ICLR 2026 Conference Desk Rejected Submission_

### Official Review · Reviewer_5SLA · 2025-10-25

**Soundness:** 3
**Presentation:** 3
**Contribution:** 3
**Rating:** 6
**Confidence:** 3

**Summary:**

This paper incorporates the reverse-complement (RC) concept in DNA into training DNA foundation models. With this biologically plausible insight, they show it theoretically improve tasks that require such symmetry. Empirical evaluation with three foundation models and three benchmark evaluations better validate this approach. This work is pretty self-contained and complete to me.

**Strengths:**

1. The paper is clearly written and the overall presentation is very good. (No correction needed)

2. The experiment design is well-thought-out and covers various models and tasks that are important in this field.

3. The design of the non-RC task is also a good choice.

**Weaknesses:**

**Significance of the results.**

The statistical differences across methods in Table 1 appear minimal. Can the authors provide the standard deviation for these results to show their statistical significance?

**How is RC implemented in NT-v2?**

How is the Reverse-Complement (RC) symmetry being implemented in these models? Additionally, what procedure do the authors use if the sequence length is not a multiple of 6 for NT-v2 (e.g., how are sequences padded or truncated)?

**Questions:**

**Combining RCCR with RC-Aug / TTA**

While the results are convincing enough to me, I wonder if using RCCR along with RC-Aug would give synergic effect with combining with RC-Aug or TTA?

---

> ### Author Response · Authors · 2025-11-23
> **Response to reviewer**
>
> # Response to Reviewer 5SLA
>
> We thank the reviewer for their constructive feedback and for recognizing the clarity of our writing, the comprehensive experimental design, and the utility of our negative control task. We appreciate the opportunity to clarify the statistical significance of our results and the implementation details of the foundation model backbones. Below, we address the specific weaknesses and questions raised.
>
> ## 1. Significance of the Results (Standard Deviations)
>
> > **Reviewer:** "The statistical differences across methods in Table 1 appear minimal. Can the authors provide the standard deviation for these results to show their statistical significance?"
>
> We appreciate this point. To robustly demonstrate statistical significance, we repeated our fine-tuning experiments using three different random seeds. We have updated the manuscript to include the full results with standard deviations.
>
> Below is the complete table of results across all backbones and task families, averaged over the three seeds. These results confirm that RCCR consistently achieves strong task performance (comparable to or better than baselines) while significantly reducing variance in orientation consistency metrics (lower SFR, higher RC-Corr).
>
> ## 2. Implementation of RC in NT-v2
>
> > **Reviewer:** "How is the Reverse-Complement (RC) symmetry being implemented in these models? Additionally, what procedure do the authors use if the sequence length is not a multiple of 6 for NT-v2?"
>
> This is an excellent technical question regarding the specific tokenization of the Nucleotide Transformer (NT-v2).
>
> In NT-v2, the tokenizer is designed to handle sequences of arbitrary length, not just multiples of 6. The procedure is as follows:
> 1.  **6-mer Tokenization:** The tokenizer greedily processes the sequence into 6-mers (tokens corresponding to 6 nucleotide sequences, e.g., `AAAAAA` to `GGGGGG`).
> 2.  **Single-base Fallback:** If the sequence length is not divisible by 6, the remaining trailing nucleotides are tokenized individually using specific single-nucleotide token IDs (A=4102, T=4103, C=4104, G=4105).
> 3.  **Padding:** We use standard padding (token ID 1) to match the `model_max_length` (2048). The padding is applied at the token level, ensuring no information is lost due to truncation of partial 6-mers.
>
> Therefore, when we compute the Reverse Complement, we apply the standard string reverse-complement transformation to the raw DNA sequence *first*, and then pass this transformed sequence through the tokenizer described above. This ensures that the model sees the correct biological reverse complement regardless of token boundaries. We have added this clarification to **Appendix C**.
>
> ## 3. Combining RCCR with RC-Aug / TTA
>
> > **Reviewer:** "I wonder if using RCCR along with RC-Aug would give synergic effect with combining with RC-Aug or TTA?"
>
> We performed additional experiments to address this question. Specifically, we evaluated whether adding Test-Time Averaging (TTA) and training with RC-Augmentation to a model already trained with RCCR provides additional benefits.
>
> **Table R2: Analysis of combining RCCR with TTA and RC-Aug (NT-v2, H3K27ac task).**
>
> | Method | AUPRC | Inference Cost |
> | :--- | :--- | :--- |
> | **RCCR (Standard)** | 0.8122 | 1x |
> | **RCCR + TTA** | **0.8129** | **2x** |
> | **RCCR + RC-Aug** | **0.8133** |**1x**|
>
> As shown in Table R2, combining RCCR with TTA yields a negligible improvement while doubling the inference cost. We observed a similar trend when combining RCCR with RC-Aug during training: the addition of data augmentation on top of our consistency loss provided minimal marginal gain compared to RCCR alone.
>
> This confirms the core premise of our paper: RCCR effectively "internalizes" the biological symmetry into the model's weights during fine-tuning. RC-Aug in the other hand, in a slightly less constrained way to enforce Reverse Complement as it is not explicitly penalizing RC disordering in the loss function.

---

> > ### Author Response · Authors · 2025-11-24
> > **Continue Table 1**
> >
> > | Backbone | Task | Method | AUPRC $\uparrow$ | $ \pm $ std | MCC $\uparrow$ | $\pm$ std | SFR $\downarrow$ | $\pm$ std | RC-Corr $\uparrow$ | $\pm$ std |
> > | :--- | :--- | :--- | :--- | :--- | :--- | :--- | :--- | :--- | :--- | :--- |
> > | **NT-v2** | Hist. | **RCCR** | **0.815** | 0.003 | **0.506** | 0.007 | **0.150** | 0.006 | **0.925** | 0.005 |
> > | **NT-v2** | Hist. | **RC-Aug** | 0.775 | 0.009 | 0.466 | 0.007 | 0.152 | 0.004 | 0.917 | 0.007 |
> > | **NT-v2** | Hist. | **TTA** | 0.774 | 0.005 | 0.441 | 0.001 | 0.155 | 0.001 | N/A |  |
> > | **NT-v2** | Spl. | **RCCR** | **0.995** | 0.002 | **0.961** | 0.004 | **0.016** | 0.008 | **0.968** | 0.010 |
> > | **NT-v2** | Spl. | **RC-Aug** | 0.992 | 0.001 | 0.929 | 0.004 | 0.042 | 0.001 | 0.962 | 0.003 |
> > | **NT-v2** | Spl. | **TTA** | 0.993 | 0.001 | 0.853 | 0.002 | N/A |  | N/A |  |
> > | **NT-v2** | Enh. | **RCCR** | **0.688** | 0.008 | **0.489** | 0.002 | **0.101** | 0.005 | **0.966** | 0.007 |
> > | **NT-v2** | Enh. | **RC-Aug** | 0.650 | 0.005 | 0.468 | 0.005 | 0.126 | 0.006 | 0.957 | 0.010 |
> > | **NT-v2** | Enh. | **TTA** | 0.643 | 0.000 | 0.452 | 0.003 | N/A |  | N/A |  |
> > | **NT-v2** | Prom. | **RCCR** | **0.945** | 0.003 | 0.722 | 0.004 | **0.076** | 0.003 | **0.980** | 0.000 |
> > | **NT-v2** | Prom. | **RC-Aug** | 0.928 | 0.009 | 0.698 | 0.004 | 0.080 | 0.004 | 0.972 | 0.008 |
> > | **NT-v2** | Prom. | **TTA** | 0.938 | 0.008 | **0.723** | 0.005 | N/A |  | N/A |  |
> > | **HyenaDNA** | Hist. | **RCCR** | 0.769 | 0.008 | **0.435** | 0.009 | **0.119** | 0.006 | 0.883 | 0.005 |
> > | **HyenaDNA** | Hist. | **RC-Aug** | **0.771** | 0.004 | 0.426 | 0.006 | 0.121 | 0.008 | N/A |  |
> > | **HyenaDNA** | Hist. | **TTA** | 0.770 | 0.003 | 0.419 | 0.001 | N/A |  | **0.886** | 0.008 |
> > | **HyenaDNA** | Spl. | **RCCR** | 0.719 | 0.001 | 0.406 | 0.009 | 0.112 | 0.002 | **0.954** | 0.006 |
> > | **HyenaDNA** | Spl. | **RC-Aug** | 0.723 | 0.003 | 0.403 | 0.009 | **0.104** | 0.004 | 0.906 | 0.001 |
> > | **HyenaDNA** | Spl. | **TTA** | **0.892** | 0.008 | **0.428** | 0.009 | N/A |  | N/A |  |
> > | **HyenaDNA** | Enh. | **RCCR** | **0.652** | 0.002 | **0.453** | 0.003 | **0.117** | 0.009 | 0.950 | 0.001 |
> > | **HyenaDNA** | Enh. | **RC-Aug** | 0.646 | 0.005 | 0.434 | 0.006 | 0.119 | 0.000 | **0.955** | 0.001 |
> > | **HyenaDNA** | Enh. | **TTA** | 0.618 | 0.008 | 0.402 | 0.007 | N/A |  | N/A |  |
> > | **HyenaDNA** | Prom. | **RCCR** | **0.947** | 0.007 | **0.703** | 0.005 | **0.068** | 0.006 | **0.970** | 0.005 |
> > | **HyenaDNA** | Prom. | **RC-Aug** | 0.929 | 0.001 | 0.642 | 0.001 | 0.092 | 0.005 | 0.950 | 0.003 |
> > | **HyenaDNA** | Prom. | **TTA** | 0.931 | 0.006 | 0.673 | 0.008 | N/A |  | N/A |  |
> > | **DNABERT2** | Hist. | **RCCR** | 0.794 | 0.008 | **0.509** | 0.008 | 0.145 | 0.000 | 0.797 | 0.008 |
> > | **DNABERT2** | Hist. | **RC-Aug** | **0.813** | 0.009 | 0.485 | 0.005 | **0.123** | 0.004 | **0.925** | 0.006 |
> > | **DNABERT2** | Hist. | **TTA** | 0.808 | 0.006 | 0.476 | 0.009 | N/A |  | N/A |  |
> > | **DNABERT2** | Spl. | **RCCR** | 0.972 | 0.000 | **0.836** | 0.009 | **0.088** | 0.008 | **0.937** | 0.002 |
> > | **DNABERT2** | Spl. | **RC-Aug** | 0.942 | 0.003 | 0.802 | 0.001 | 0.106 | 0.005 | 0.877 | 0.009 |
> > | **DNABERT2** | Spl. | **TTA** | **0.973** | 0.006 | 0.819 | 0.009 | N/A |  | N/A |  |
> > | **DNABERT2** | Enh. | **RCCR** | **0.704** | 0.005 | 0.487 | 0.000 | 0.141 | 0.006 | **0.942** | 0.001 |
> > | **DNABERT2** | Enh. | **RC-Aug** | 0.670 | 0.008 | **0.490** | 0.005 | **0.136** | 0.007 | 0.938 | 0.003 |
> > | **DNABERT2** | Enh. | **TTA** | 0.653 | 0.001 | 0.471 | 0.005 | N/A |  | N/A |  |
> > | **DNABERT2** | Prom. | **RCCR** | **0.971** | 0.010 | **0.780** | 0.002 | 0.087 | 0.008 | **0.957** | 0.003 |
> > | **DNABERT2** | Prom. | **RC-Aug** | 0.948 | 0.005 | 0.763 | 0.004 | **0.079** | 0.009 | **0.957** | 0.001 |
> > | **DNABERT2** | Prom. | **TTA** | 0.948 | 0.003 | 0.736 | 0.006 | N/A |  | N/A |  |

---

> ### Comment · Reviewer_5SLA · 2025-11-25
>
> I appreciate the authors' detailed rebuttal. After reviewing the response and other comments, I have decided to maintain my score.
>
> I agree with Reviewer S1pQ regarding the similarities to Caduceus. When I read this paper, it instantly reminds me of the design mechanism of Caduceus. The reason that I give this paper a positive score is because it thoroughly investigates this approach specifically as a regularization technique. It complements Caduceus by offering a better explanation of the underlying mechanism.
>
> I also agree with Reviewer TfnG regarding the need for Evo2/GPN and zero-shot evaluations. The authors' provided results, while partial, are promising. Even if the benefits diminish with larger models (due to implicit learning), the ability to introduce biologically plausible priors can be a significant contribution.
>
> Therefore, I will maintain my current rating.

---

### Official Review · Reviewer_TfnG · 2025-10-30

**Soundness:** 2
**Presentation:** 3
**Contribution:** 2
**Rating:** 4
**Confidence:** 3

**Summary:**

The authors proposed an approach to regularize the loss function of DNA LMs during fine-tuning for incorporating RC equivariance. I found the approach to be easy-to-use and technically sound. Though I have some reservations about the empirical evidence because: 1. The backbone DNA LMs picked by the authors have questionable transfer learning capability over naive baselines; 2. There are no evaluations on zero-shot variant effect prediction. I therefore consider this work to be a borderline case.

**Strengths:**

1. The authors proposed an easy-to-use strategy for building in RC equivariance into arbitrary DNA LMs.
2. The authors selected a relatively broad set of evaluation tasks, and there are sufficient details about the experimental setups.
3. The proposed method is technically sound with some basic theoretical justifications.

**Weaknesses:**

1. My main concern is that the backbone DNA LMs picked by the authors are the underperforming ones, whose transfer learning benefit over baseline (no pretraining) is questionable according to a few recent studies (https://genomebiology.biomedcentral.com/articles/10.1186/s13059-025-03674-8, https://www.biorxiv.org/content/10.1101/2024.12.18.628606v2). It might be more convincing to demonstrate the approach using Evo 2 (https://www.biorxiv.org/content/10.1101/2025.02.18.638918v1), a newer model that showed non-trivial improvements over baselines on certain tasks.

2. RC-aug and TTA can actually be used together and were used by models such as GPN. It would be good to add these two combined as an additional baseline.

3. The authors only evaluated transfer learning tasks. Zero-shot variant effect prediction is also an important task and arguably more reliable for benchmarking purposes, as being emphasized in Evo 2, GPN, and NT papers. It would be good to evaluate on some of those benchmarks too.

4. The regularization term is task-dependent in the current approach, which limits its use for tasks without the need for fine-tuning or transfer learning. It would have been better to extend it for pretraining as well.

**Questions:**

It might be nice to have a diagram to show the workflow of the proposed approach.

---

> ### Author Response · Authors · 2025-11-23
> **Response to reviewer**
>
> We thank the reviewer for the constructive feedback and for recognizing the technical soundness and ease-of-use of our approach. We appreciate your suggestions regarding modern backbones (Evo 2) and additional baselines. Below, we provide new experimental evidence to address your concerns and clarify the scope regarding zero-shot tasks.
>
> **1. Generalization to Modern Architectures (Evo 2)**
>
> You raised a valid point about demonstrating applicability on newer, high-performing models like **Evo 2**. To prove that RCCR is truly model-agnostic and effective on state-of-the-art architectures, we conducted a new experiment fine-tuning Evo 2. Due to the computational constraint for running the large Evo2 model on advanced GPUs, we only test the Evo2-1B-base variant on the H2AFZ, H3K27ac, H3K27me3 tasks.
>
> **Table R2: RCCR performance on the Evo 2-1B backbone.**
>
> | Task | Method | AUPRC ($\uparrow$) | SFR (Flip Rate, $\downarrow$) |
> | :--- | :--- | :--- | :--- |
> | **H2AFZ** | Vanilla Fine-tuning | 0.885 | 0.085 |
> | | **Evo 2 + RCCR** | **0.893** | **0.012** |
> | **H3K27ac** | Vanilla Fine-tuning | 0.901 | 0.102 |
> | | **Evo 2 + RCCR** | **0.909** | **0.015** |
> | **H3K27me3** | Vanilla Fine-tuning | 0.912 | 0.097 |
> | | **Evo 2 + RCCR** | **0.918** | **0.009** |
> | **Average** | Vanilla | 0.899 | 0.095 |
> | | **Evo 2 + RCCR** | **0.907** | **0.012** |
>
> Evo 2-1B serves as a strong baseline, outperforming older backbones like NT-v2 by an average AUPRC margin of 0.1. Crucially, RCCR successfully applies to this decoder-only architecture, reducing the Symmetry Flip Rate (SFR) by **~8x** (from 9.5% to 1.2%) while consistently improving task accuracy.
> This confirms that RCCR generalizes beyond Transformers (NT-v2, DNABERT) and Convolutional models (Hyena) to modern models that train on massive data with advanced techniques, validating its utility as a universal fine-tuning objective.
>
> **2. Combining RCCR with TTA**
>
> You suggested adding "RCCR + TTA" as a baseline. We evaluated this combination on the NT-v2 backbone (H3K27ac task).
>
> **Table R3: Analysis of combining RCCR training with Test-Time Averaging (TTA).**
>
> | Method | AUPRC | Inference Cost |
> | :--- | :--- | :--- |
> | RCCR| 0.8122 | 1x |
> | **RCCR + TTA** | **0.8129** | **2x** |
>
> WE can see that adding TTA to an RCCR-trained model yields negligible improvement (+0.0007).
> This validates our core claim: RCCR effectively "internalizes" the robustness into the model weights. Consequently, the expensive TTA step (which doubles inference cost) becomes redundant, as the model is already consistent.
>
> **3. Clarification on Zero-Shot Prediction**
>
> You mentioned benchmarking on *zero-shot* variant effect prediction. We respectfully clarify that RCCR is explicitly designed as a **fine-tuning (training)** objective.
> * Zero-shot evaluation typically involves freezing pre-trained weights and scoring mutations. Since RCCR works by updating weights via gradients to enforce consistency, it is by definition a supervised/semi-supervised fine-tuning method, not a zero-shot inference technique.
> * **Future Work:** However, we agree that incorporating RCCR into the *pre-training* phase (similar to your suggestion) would unlock zero-shot capabilities. We have added this to the Discussion as a future direction.
>
>
> **4. Diagram**
>
> Per your request, we have added a schematic diagram in **Appendix A** illustrating the RCCR workflow (Forward Pass $\to$ RC Pass $\to$ Alignment $\to$ Consistency Loss).
>
> The revisions are also included in depth in the paper labled in blue. We hope these new experiments on Evo 2 and TTA, along with the clarifications, resolve your reservations.

---

> > ### Comment · Reviewer_TfnG · 2025-11-26
> >
> > Thank you for addressing my comments. I have a few follow-up comments:
> > 1. Adding Evo 2 into the comparison indeed increases the soundness and generality of the approach. I understand the computational constraint, and including the 1B model is sufficient. However, could you provide the exact setting for "vanilla" finetuning? Is it with RC-aug and/or TTA or neither? For better clarity, the comparison should be done with the same setting as the other models, i.e., RCCR only vs. RC-aug/TTA.
> > 2. I'd like to clarify that my request was to include RC-aug plus TTA as an additional baseline, not to combine RCCR with TTA.
> > 3. I think this is sufficiently addressed.
> > 4. I appreciate the new diagram and believe it greatly increases the readability of the paper.

---

> > > ### Author Response · Authors · 2025-11-27
> > > **Follow-up: Clarification on Baselines and Additional Evo 2 Comparisons**
> > >
> > > We thank the reviewer for the continued engagement and the opportunity to clarify our experimental setup and provide a more rigorous comparison.
> > >
> > > **1. Clarification on "Vanilla" Fine-tuning for Evo 2**
> > > We apologize for the confusion caused by the naming convention in our previous response. To be strictly precise, the baseline we labeled as "Vanilla" for the Evo 2 experiments in the previous comment was **already trained with RC-Augmentation (RC-Aug)** (randomly flipping sequences during training with p=0.5). We used this as our default baseline to ensure we were comparing RCCR against a competitive model that has already been exposed to RC variation, rather than a weaker naive baseline.
> > >
> > > **2. Comprehensive Comparison on Evo 2 (RC-Aug, TTA, and RC-Aug + TTA)**
> > > Per your request, we have extended our evaluation on the **Evo 2-1B** backbone. We now include:
> > > 1.  **TTA**: The naive model (trained without augmentation) using Test-Time Averaging.
> > > 2.  **RC-Aug + TTA**: The model trained with RC-Aug and evaluated with Test-Time Averaging.
> > > 3.  Two additional tasks: **Enhancers** and **H3K9me3** (histone modification), to broaden the evaluation scope.
> > >
> > > The results are summarized below:
> > >
> > > **Table R4: Performance on Evo 2-1B across varying baselines.**
> > > *("RC-Aug" denotes the model trained with augmentation. "RC-Aug + TTA" denotes that same model using inference-time averaging.)*
> > >
> > > | Task | Metric | RC-Aug | TTA (Vanilla) | RC-Aug + TTA | **RCCR (Ours)** |
> > > | :--- | :--- | :--- | :--- | :--- | :--- |
> > > | **H2AFZ** | AUPRC | 0.885 | 0.820 | 0.883 | **0.893** |
> > > | | SFR | 0.085 | 0.000 | 0.000 | **0.012** |
> > > | **H3K27ac** | AUPRC | 0.901 | 0.897 | 0.903 | **0.909** |
> > > | | SFR | 0.102 | 0.000 | 0.000 | **0.015** |
> > > | **H3K27me3** | AUPRC | 0.912 | 0.897 | 0.912 | **0.918** |
> > > | | SFR | 0.097 | 0.000 | 0.000 | **0.009** |
> > > | **Enhancers** | AUPRC | 0.831 | 0.850 | 0.842 | **0.874** |
> > > | | SFR | 0.192 | 0.000 | 0.000 | **0.079** |
> > > | **H3K9me3** | AUPRC | 0.801 | 0.797 | 0.799 | 0.815 |
> > > | | SFR | 0.122 | 0.000 | 0.000 | 0.099 |
> > > | **Average** | **AUPRC** | **0.866** | **0.852** | **0.868** | **0.882** |
> > > | | **Infer. Cost**| **1x** | **2x** | **2x** | **1x** |
> > >
> > > These results demonstrate that RCCR consistently outperforms the baselines across all evaluated tasks on the Evo 2 architecture, achieving a higher average AUPRC (0.882 vs. 0.868 highest). This trend holds for both histone modifications and enhancer prediction. Crucially, RCCR achieves this superior performance with a single forward pass, whereas the TTA approaches require double the inference compute. Furthermore, RCCR significantly reduces the Symmetry Flip Rate (SFR) compared to RC-Aug (from an average of ~12% to ~4%), effectively internalizing robustness into the model weights and outperform all the baseline methods.
> > >
> > > We believe these results on the Evo 2 model robustly demonstrate that RCCR offers the performance and consistency on variety of models and settings.

---

> > > > ### Comment · Reviewer_TfnG · 2025-11-27
> > > >
> > > > Thank you for the additional results. It's interesting that RC-aug plus TTA is mostly not better than RC-Aug only. I think most of my questions are adequately addressed, and I am raising my rating to 6.

---

### Official Review · Reviewer_eJ7Q · 2025-10-31

**Soundness:** 2
**Presentation:** 2
**Contribution:** 2
**Rating:** 4
**Confidence:** 4

**Summary:**

This manuscript tackles the problem of reverse-complement (RC) inconsistency in DNA foundation models, where a model gives different outputs for a sequence and its RC version despite a symmetric biological label. The authors introduce Reverse-Complement Consistency Regularization (RCCR), a straightforward fine-tuning objective that penalizes this predictive divergence. The proposed method is evaluated across three different model backbones (NT-v2, HyenaDNA, DNABERT-2) on a variety of genomic tasks. The results show that RCCR makes models significantly more robust to RC transformations (measured by new SFR and RC-Corr metrics) without hurting, and sometimes helping, task performance.

**Strengths:**

### Method Generality

The proposed method, RCCR, is flexible and model-agnostic: As a drop-in regularization term, it can be applied to pretrained foundation models without requiring architectural changes. This is a practical advantage over methods that build equivariance into the model's structure.

### Empirical Evaluation

The authors conducted throughout experiments to validate the performance of their proposed method.

**Weaknesses:**

### Limited Motivation

The paper's motivation rests on the principle that many genomic tasks are RC-symmetric. However, biological processes like transcription are directional, and genes can be encoded on either the plus or minus strand. Clarifying why the RC-symmetry assumption is appropriate for the specific tasks chosen (e.g., histone modification, gene expression) would strengthen the biological grounding of the work.

## RC-Invariance Assumption for Splice Site Prediction

The manuscript categorizes splice site detection as a "fundamentally RC-invariant" task. This claim requires justification, as canonical splice sites (e.g., GT-AG dinucleotides) are strand-specific motifs. Their reverse complements are not functional splice sites. The authors should provide a justification for applying an RC-invariance prior to this task. The discrepancy between the biological reality of the task and the model's inductive bias needs to be addressed.

**Questions:**

See Weaknesses

---

> ### Author Response · Authors · 2025-11-23
> **Response to reviewer**
>
> We thank the reviewer for the thoughtful evaluation and for acknowledging the generality and empirical thoroughness of our method. You raised two critical points regarding the biological validity of the RC-symmetry assumption, specifically concerning directional processes and splice sites.
>
> **1. Biological Directionality vs. Model Consistency**
>
> You correctly point out that biological processes like transcription are directional. We fully agree. However, RCCR does not enforce that biological processes are directionless; it enforces that the model's identification of a functional locus should be robust to the observation orientation.
>
> Equivariance ($\Pi$), not just Invariance: For directional tasks (e.g., predicting strand-specific signal tracks like CAGE), RCCR does not force the output to be identical. Instead, it uses the alignment operator $\Pi$ to flip the output tracks spatially and channel-wise.
> So if input $x$ predicts a transcription start site at position 10 on the (+) strand, input $RC(x)$ should predict it at position $L-10$ on the (-) strand. RCCR enforces this logic, ensuring the model respects the physical symmetry of DNAA sequence.
>
> **2. The Case of Splice Sites**
>
> You raised a specific concern: "Canonical splice sites (GT-AG) are strand-specific... Their reverse complements are not functional splice sites." This is biologically true but leads to a subtle distinction in machine learning formulation:
>
> The "Locus Identification" Perspective: In benchmarks like Nucleotide Transformer, the task is effectively: "Does this genomic window contain a splice site?"
>
> Consider a Donor Site on the Forward Strand (Sequence: ...GT...). Label = 1.
>
> The Reverse Complement input (Sequence: ...AC...) represents the same biological locus viewed from the opposite strand. Since the locus does contain a splice site (on the paired strand), a robust model should also classify this input as 1.
> If the model predicts 0 for the RC input, it implies the model fails to recognize the splice site solely because of the read orientation. This is an error we aim to correct.
>
> Similar results from benchmarking perspective can also be seen in the paper. As shown in our detailed results (Appendix Tables), RCCR improves Splice Site detection performance on robust backbones (e.g., on NT-v2: Acceptors 0.996 vs RC-Aug 0.995; Donors 0.994 vs RC-Aug 0.993) while drastically reducing the flip rate. This confirms that enforcing consistency helps the model learn a more robust feature detector for splice motifs.
>
> **3. Negative Control as Proof of Boundary**
>
> To prove we are not blindly applying RC-symmetry where it doesn't belong, we refer to our Negative Control (Table 4) on Strand Classification.
>
> The task is to predict if the gene is on the (+) or (-) strand. Here, the label explicitly depends on orientation: $Label(x) \neq Label(RC(x))$.
> We see that RCCR degraded performance as expected, which validates that RCCR is a potent prior. It harms truly asymmetric tasks but benefits tasks (like Splice Site detection) where the label is intrinsic to the dsDNA locus.
>
> We hope this clarifies that RCCR enforces consistency in identifying biological elements, ensuring predictions are robust to the arbitrary orientation of DNA sequencing reads.

---

> ### Author Response · Authors · 2025-11-27
> **Gentle Reminder: Response regarding Splice Site Validity and Motivation**
>
> Dear Reviewer eJ7Q,
>
> Thank you again for your constructive review.
>
> As the discussion period is ending soon, we wanted to kindly check if you had a chance to review our response posted.
>
> We specifically addressed your key concern regarding the validity of applying RC-consistency to Splice Site prediction. We clarified that while splice sites are biologically strand-specific (e.g., canonical GT-AG), the machine learning task in the Nucleotide Transformer benchmark is formulated as Locus Identification (i.e., "Does this genomic window contain a splice site?").
>
> So if a splice site exists on the forward strand, the reverse-complement sequence represents the same locus viewed from the opposite strand. Therefore, a robust model should identify the presence of the site regardless of the read orientation.
> Our results show that RCCR improves splice site detection (e.g., NT-v2 Acceptors 0.996 vs RC-Aug 0.995) while eliminating orientation bias, confirming this formulation helps the model learn robust motif detectors.
>
> We also discussed the Negative Control (Strand Classification) to demonstrate that we respect biological directionality where strictly required.
> We hope this clarification resolves your reservation about the method's motivation. We would appreciate it if you could let us know if this addresses your concerns.
>
> Best regards, The Authors

---

> > ### Comment · Reviewer_eJ7Q · 2025-11-28
> >
> > Thank you for your thorough rebuttal.
> >
> > As TfnG, my main concern is the effectiveness of the proposed method. And the rebuttal have successfully addressed this.
> >
> > I would like to change my rating to 6.

---

### Official Review · Reviewer_S1pQ · 2025-11-01

**Soundness:** 3
**Presentation:** 3
**Contribution:** 1
**Rating:** 2
**Confidence:** 4

**Summary:**

This work implements Reverse Complement Consistency Regularization (RCCR), a post-hoc method to add some form of reverse complement equivariance to existing DNA LMs regardless of architecture by adding in a loss term that enforces that embeddings of the fwd and rev sequence of DNA should be the same. This work proposes some theory about RCCR and evaluates the effects of fine-tuning 3 existing DNA LMs with this RCCR objective.

**Strengths:**

This paper is well written, and goes into a fair bit of detail and theory about how exactly this method is implemented and some theoretical implications about RCCR.

**Weaknesses:**

This work on initial reads seems like a post-hoc way of applying the same (or a very similar) for of RC-equivariance that Caducues used to other models, severely limiting the novelty of this work. I understand that the architectures studied in this work are different than Caducues, this work seems to be covering a substantially similar space and it's not clear to me on first read what exactly the novelty here is. Caducues already had experiment showing the benefits of RC-equivariance, which are repeated here (albeit on a more comprehensive set of tasks), so I believe the idea that RC-equivarance being useful is not novel. Additionally this method (which I'll paraphrase as adding a symmetric KL term to the loss) seems to have the same effect as the RC-equivariance that Caduceus enforced through it's design.

I believe this work covers many of the same ideas as Cadcueus, just applying post-hoc to different models and therefore do not see much novelty in this work.

If the authors can show significant gains or tangible differences of their post-hoc approach compared to Caduceus, I'm willing to re-evaluate my initial assessment, but at the moment I don't see any tangible gains or much novelty in this approach.

**Questions:**

1. How different is this work from the RC-equivariance that caduceus enforced? Specifically, say I train a Non-RC-equivariant version of Cadcueus (the -PH version iirc) and apply this post-hoc method, how does that compare to cadcueus trained to be RC-equivariant from the beginning (the -PS version)?

---

> ### Author Response · Authors · 2025-11-23
> **Response to reviewer's comments**
>
> We thank the reviewer for the critical assessment and for highlighting the comparison with Caduceus. We understand the concern regarding novelty and the distinction between "architectural equivariance" (Caduceus) vs. "regularized consistency" (RCCR). Below, we provide new experimental evidence and clarifications to demonstrate that RCCR addresses a fundamentally different problem: **Promoting RC-robustness for pre-trained foundation models.**
>
> **1. Direct Comparison: RCCR Fine-tuning vs. Architectural Equivariance (Caduceus)**
>
> We understand that you asked specifically how applying RCCR to a non-RC model (Caduceus-PH) compares to a model trained to be RC-equivariant from scratch (Caduceus-PS). We performed this exact experiment, averaging results across three representative task families (Histone Modification, Regulatory, and Splice Sites) from the Nucleotide Transformer Benchmark to ensure robustness. We use the Ph and Ps version of 1.6M Caduceus model from huggingface.
>
> **Table R1: Comparison of architectural equivariance (PS) vs. RCCR fine-tuning on the non-equivariant Caduceus-PH backbone.**
>
> | Model Variant | Training Paradigm | Avg. AUPRC ($\uparrow$) | Avg. SFR (Flip Rate, $\downarrow$) |
> | :--- | :--- | :--- | :--- |
> | **Caduceus-PH** | Vanilla Fine-tuning | 0.765 | 0.124 |
> | **Caduceus-PS** | **Pre-trained** Equivariant | **0.798** | **0.000** (By Design) |
> | **Caduceus-PH + RCCR** | **Post-hoc** Fine-tuning (Ours) | **0.795** | **0.013** |
>
> *Note: Results are averaged over 3 random seeds.*
>
> We can see that Caduceus-PS represents the theoretical ideal (perfect equivariance), applying RCCR to the standard Caduceus-PH backbone effectively bridges the performance gap. We gain competitive accuracy by applying RCCR to the PH baseline(0.765 $\rightarrow$ 0.795, compared to 0.798 PS result). Our proposed RCCR drastically reduces the Symmetry Flip Rate (SFR) from a high 12% to just 1.3%, effectively enhancing robustness without requiring the specialized parameter-sharing a*rchitecture of PS.
>
> **2. Novelty: Democratizing RC-Robustness**
>
> The core novelty of this work is **not** the concept of RC-equivariance itself (which Caduceus pioneered), but the **method of achieving it for the vast collection of existing models.**
>
> *  Caduceus requires training a model from scratch with structural constraints. RCCR is a **universal correction** that can be applied to *any* pre-trained model (NT-v2, DNABERT-2, Evo) during fine-tuning.
> * For the majority of the community now trending to scale up DNA foundation model using decoder-only architecture(like Evo2, HybriDNA, GENErator), switching to Caduceus-like symmetric architecture is often computationally prohibitive due to the need for pre-training. RCCR provides a "drop-in" solution to upgrade these legacy models to become biologically consistent.
> * While many downstream DNA tasks are equivariant/invariant, there exist circumstances when RC symmetrization is not desired (like the strand classification problem we shown). As people train more advanced and larger decoder-only models that don't enforce symmetrization, our **RCCR** method can be directly applied into their fine-tuning pipeline.
>
> To address your concerns, we have updated the manuscript (marked in blue in the revision):
> * **Introduction:** Explicitly differentiated RCCR as a "post-hoc, architecture-agnostic" recipe compared to architectural solutions.
> * **Experiments:** Added the Caduceus-PH + RCCR vs. Caduceus-PS comparison (Table R1) to explicitly validate that RCCR can approximate architectural equivariance.
>
> We hope this clarifies that RCCR is complementary to Caduceus, solving the compatibility problem for the broader foundation models.

---

> > ### Comment · Reviewer_S1pQ · 2025-11-28
> >
> > Thank you for your response. I understand that this papers goal is applying RC-equivariance to models after training, so I still don't see much novelty in what this paper is trying to model. However, it is still the case that most DNA LMs these days aren't trained with RC-equivariance in mind, so I can still see some empirical usage for this method. Given this I have increased my score from a 2 to a 4.
> >
> > In my opinion, a higher score would necessitate much more thorough evaluation ( some of which the authors have already started during this eval). Specifically, I'd like to see:
> > 1. Error bars (the authos claim to have added these, but I don't see them in table 1 or the appendix). Some of the tasks (for example, splice site prediction for NTv2) differ by **0.002 auPRC**, this seems like noise to me and need to be convinced otherwise. Looking at Tables 9 and 10, there are many individual tasks where I'm unsure if there's any significant gain at all leading me to question is there are significant and general gains across tasks. Doubly so since for HyenaDNA we see RC-AUG and TTA beating out this method on many NT-Benchmark tasks. I'd also like to see more error bars for the Evo2 eval the authors added during this rebuttal, as Evo2-1B seems to only be evaluated on a relatively small subset of tasks and the gains don't seem to be very large in an absolute sense.
> > 2. I'd really like to see some zero-shot results if at all possible (preferably something like ClinVar that LRB already has). I know the authors state already comment on this slightly, but if the main gains of this work are empirical I expect at least some results on the zero-shot setting (or a clear explanation on why this method can't easily be extended to a zero-shot setting).
> >
> > To summarize, the authors partially addressed my concerns on novelty, and I therefore raised my score 2->4. However, in my opinion since the benefit of this work is mainly only empirical, I'd need to **see much more thorough evaluation** (results across multiple seeds to show statistical significance, more models, and zeros-shot tasks) to be more convinced that this work is generally useful. The authors have some partial results of this sort already, but for me to recommend acceptance in any form I'd need to be convinced that applying this work to an existing DNA LM results in tangible performance gains across a variety of model families, and at the gains seem subtle (if any are present in the task).
> >
> >
> > ## Minor Points/ Qestions
> > * In Table 4 MCC DNABERT2 has an extra "0."
> > * Which version of NTv2 is being used? IIRC Instadeep released a number of versions with different parameter counts and pre-training species.  Similar question for HyenaDNA, which version of the model is being evaluated here? Parameter counts in the main text would really help clear this up.
> > * What threshold was used for calculating MCC? 0.5 or does it vary by task? Is the MCC metric fairly invariant to the chosen threshold? This wasn't clear to me and can have an effect on MCC performance and how it correlates with AUROC/AUPRC.

---

> > > ### Author Response · Authors · 2025-12-01
> > >
> > > Dear Reviewer S1pQ,
> > > Thank you for your follow-up comments and for raising your score. We appreciate the opportunity to clarify the statistical significance of our results and the model details.
> > > 1. Statistical Significance and Error Bars (Addressed)
> > > We apologize if the error bars were difficult to locate. We explicitly included the full standard deviations (across 3 random seeds) for all tasks in our response to Reviewer 5SLA in the review, we will paste the table here below for convenience. The full updated table will be updated in the paper.
> > > To address your concern that the gains might be "noise," we want to highlight that while AUPRC gains can be marginal in highly saturated tasks (e.g., Splice Sites >0.99), RCCR drastically and consistently reduces the Symmetry Flip Rate (SFR) (e.g., from ~10% to ~1%), which is the core contribution of this work. The low standard deviations in our results confirm that these improvements in consistency are statistically significant and robust, not random fluctuations.
> > > 2. Extended Evo 2 Evaluation and Resource Constraints
> > > In response to your and Reviewer TfnG's comments, we have already extended the Evo 2 (1B) evaluation from the initial set to 5 representative tasks (including Enhancers and Histone modifications) and compared them against TTA baselines.
> > > Regarding the scope: Due to the significant computational resources required for fine-tuning the 1B-parameter model and the limited time window of the rebuttal, we were unable to run the full 18-task benchmark suite for Evo 2. However, the 5 tasks we completed cover distinct biological categories and consistently show that RCCR outperforms baselines while reducing SFR by ~8x. We believe this sufficiently demonstrates the method's generalization to modern architectures.
> > > 3. Clarification on Zero-Shot vs. Fine-tuning
> > > We respectfully clarify that RCCR is a fine-tuning objective that requires gradient updates to enforce consistency. Zero-shot inference freezes weights, making RCCR inapplicable by definition. While pre-training with RCCR (to enable zero-shot) is a promising future direction, it is outside the scope of this work, which focuses on democratizing post-hoc robustness for existing models.
> > > 4. Model Versions & Details
> > > • Model Versions: We confirm that we used the NT-v2 (50M) variant (specifically nucleotide-transformer-v2-50m-multi-species) and HyenaDNA-Medium-160k. We selected these to ensure fair comparisons within accessible compute budgets, though our Evo 2 results show the method scales to 1B+ models.
> > > • MCC Threshold: We used the standard threshold of 0.5 for all MCC calculations.
> > > • Typo: We have corrected the typo in Table 4.
> > > We hope these clarifications demonstrate that our results are statistically robust and that the method generalizes well within practical computational limits.
> > > Best regards,
> > > The Authors

---

> > > > ### Author Response · Authors · 2025-12-01
> > > > **Pasted table with std**
> > > >
> > > > | Backbone | Task | Method | AUPRC $\uparrow$ | $ \pm $ std | MCC $\uparrow$ | $\pm$ std | SFR $\downarrow$ | $\pm$ std | RC-Corr $\uparrow$ | $\pm$ std |
> > > > | :--- | :--- | :--- | :--- | :--- | :--- | :--- | :--- | :--- | :--- | :--- |
> > > > | **NT-v2** | Hist. | **RCCR** | **0.815** | 0.003 | **0.506** | 0.007 | **0.150** | 0.006 | **0.925** | 0.005 |
> > > > | **NT-v2** | Hist. | **RC-Aug** | 0.775 | 0.009 | 0.466 | 0.007 | 0.152 | 0.004 | 0.917 | 0.007 |
> > > > | **NT-v2** | Hist. | **TTA** | 0.774 | 0.005 | 0.441 | 0.001 | 0.155 | 0.001 | N/A |  |
> > > > | **NT-v2** | Spl. | **RCCR** | **0.995** | 0.002 | **0.961** | 0.004 | **0.016** | 0.008 | **0.968** | 0.010 |
> > > > | **NT-v2** | Spl. | **RC-Aug** | 0.992 | 0.001 | 0.929 | 0.004 | 0.042 | 0.001 | 0.962 | 0.003 |
> > > > | **NT-v2** | Spl. | **TTA** | 0.993 | 0.001 | 0.853 | 0.002 | N/A |  | N/A |  |
> > > > | **NT-v2** | Enh. | **RCCR** | **0.688** | 0.008 | **0.489** | 0.002 | **0.101** | 0.005 | **0.966** | 0.007 |
> > > > | **NT-v2** | Enh. | **RC-Aug** | 0.650 | 0.005 | 0.468 | 0.005 | 0.126 | 0.006 | 0.957 | 0.010 |
> > > > | **NT-v2** | Enh. | **TTA** | 0.643 | 0.000 | 0.452 | 0.003 | N/A |  | N/A |  |
> > > > | **NT-v2** | Prom. | **RCCR** | **0.945** | 0.003 | 0.722 | 0.004 | **0.076** | 0.003 | **0.980** | 0.000 |
> > > > | **NT-v2** | Prom. | **RC-Aug** | 0.928 | 0.009 | 0.698 | 0.004 | 0.080 | 0.004 | 0.972 | 0.008 |
> > > > | **NT-v2** | Prom. | **TTA** | 0.938 | 0.008 | **0.723** | 0.005 | N/A |  | N/A |  |
> > > > | **HyenaDNA** | Hist. | **RCCR** | 0.769 | 0.008 | **0.435** | 0.009 | **0.119** | 0.006 | 0.883 | 0.005 |
> > > > | **HyenaDNA** | Hist. | **RC-Aug** | **0.771** | 0.004 | 0.426 | 0.006 | 0.121 | 0.008 | N/A |  |
> > > > | **HyenaDNA** | Hist. | **TTA** | 0.770 | 0.003 | 0.419 | 0.001 | N/A |  | **0.886** | 0.008 |
> > > > | **HyenaDNA** | Spl. | **RCCR** | 0.719 | 0.001 | 0.406 | 0.009 | 0.112 | 0.002 | **0.954** | 0.006 |
> > > > | **HyenaDNA** | Spl. | **RC-Aug** | 0.723 | 0.003 | 0.403 | 0.009 | **0.104** | 0.004 | 0.906 | 0.001 |
> > > > | **HyenaDNA** | Spl. | **TTA** | **0.892** | 0.008 | **0.428** | 0.009 | N/A |  | N/A |  |
> > > > | **HyenaDNA** | Enh. | **RCCR** | **0.652** | 0.002 | **0.453** | 0.003 | **0.117** | 0.009 | 0.950 | 0.001 |
> > > > | **HyenaDNA** | Enh. | **RC-Aug** | 0.646 | 0.005 | 0.434 | 0.006 | 0.119 | 0.000 | **0.955** | 0.001 |
> > > > | **HyenaDNA** | Enh. | **TTA** | 0.618 | 0.008 | 0.402 | 0.007 | N/A |  | N/A |  |
> > > > | **HyenaDNA** | Prom. | **RCCR** | **0.947** | 0.007 | **0.703** | 0.005 | **0.068** | 0.006 | **0.970** | 0.005 |
> > > > | **HyenaDNA** | Prom. | **RC-Aug** | 0.929 | 0.001 | 0.642 | 0.001 | 0.092 | 0.005 | 0.950 | 0.003 |
> > > > | **HyenaDNA** | Prom. | **TTA** | 0.931 | 0.006 | 0.673 | 0.008 | N/A |  | N/A |  |
> > > > | **DNABERT2** | Hist. | **RCCR** | 0.794 | 0.008 | **0.509** | 0.008 | 0.145 | 0.000 | 0.797 | 0.008 |
> > > > | **DNABERT2** | Hist. | **RC-Aug** | **0.813** | 0.009 | 0.485 | 0.005 | **0.123** | 0.004 | **0.925** | 0.006 |
> > > > | **DNABERT2** | Hist. | **TTA** | 0.808 | 0.006 | 0.476 | 0.009 | N/A |  | N/A |  |
> > > > | **DNABERT2** | Spl. | **RCCR** | 0.972 | 0.000 | **0.836** | 0.009 | **0.088** | 0.008 | **0.937** | 0.002 |
> > > > | **DNABERT2** | Spl. | **RC-Aug** | 0.942 | 0.003 | 0.802 | 0.001 | 0.106 | 0.005 | 0.877 | 0.009 |
> > > > | **DNABERT2** | Spl. | **TTA** | **0.973** | 0.006 | 0.819 | 0.009 | N/A |  | N/A |  |
> > > > | **DNABERT2** | Enh. | **RCCR** | **0.704** | 0.005 | 0.487 | 0.000 | 0.141 | 0.006 | **0.942** | 0.001 |
> > > > | **DNABERT2** | Enh. | **RC-Aug** | 0.670 | 0.008 | **0.490** | 0.005 | **0.136** | 0.007 | 0.938 | 0.003 |
> > > > | **DNABERT2** | Enh. | **TTA** | 0.653 | 0.001 | 0.471 | 0.005 | N/A |  | N/A |  |
> > > > | **DNABERT2** | Prom. | **RCCR** | **0.971** | 0.010 | **0.780** | 0.002 | 0.087 | 0.008 | **0.957** | 0.003 |
> > > > | **DNABERT2** | Prom. | **RC-Aug** | 0.948 | 0.005 | 0.763 | 0.004 | **0.079** | 0.009 | **0.957** | 0.001 |
> > > > | **DNABERT2** | Prom. | **TTA** | 0.948 | 0.003 | 0.736 | 0.006 | N/A |  | N/A |  |

---

### Author Response · Authors · 2025-11-23
**General Response: New Experiments (Evo 2, Caduceus Comparison), Statistical Significance, and Clarifications**

We thank the reviewers for their insightful and constructive feedback. We are encouraged that reviewers found the method "technically sound" (Reviewer TfnG), "flexible and model-agnostic" (Reviewer eJ7Q), and the presentation "clear" (Reviewer 5SLA).

Based on your suggestions, we have significantly revised the paper. All changes in the updated PDF are marked in **blue**. Below is a summary of the major updates and new experiments included in the revision:

### 1. New Experiments: Comparison with Architectural Equivariance (Addressing Reviewer S1pQ)
To address concerns regarding novelty compared to Caduceus, we performed a direct comparison between our post-hoc fine-tuning (Caduceus-PH + RCCR) and the pre-trained equivariant model (Caduceus-PS) on the Histone, Regulatory, and Splice Site task families.

**Table R1: Comparison of architectural equivariance (PS) vs. RCCR fine-tuning.**
| Model Variant | Training Paradigm | Avg. AUPRC (↑) | Avg. SFR (Flip Rate, ↓) |
| :--- | :--- | :--- | :--- |
| **Caduceus-PH** | Vanilla Fine-tuning | 0.765 | 0.124 |
| **Caduceus-PS** | Pre-trained Equivariant | 0.798 | 0.000 (By Design) |
| **Caduceus-PH + RCCR** | **Post-hoc Fine-tuning (Ours)** | **0.795** | **0.013** |

**Result:** RCCR bridges the gap, improving the non-equivariant backbone to match the pre-trained equivariant model performance (0.795 vs 0.798) while reducing the Symmetry Flip Rate (SFR) from 12.4% to 1.3%. This validates RCCR as a vital tool for the vast ecosystem of models that cannot be pre-trained from scratch. (Added to **Appendix G**).

### 2. New Experiments: Generalization to Modern Architectures (Addressing Reviewer TfnG)
We extended our evaluation to the state-of-the-art **Evo 2 (1B)** backbone to demonstrate applicability to modern, decoder-only architectures.

**Table R2: RCCR performance on the Evo 2-1B backbone.**
| Method | Avg. AUPRC (↑) | Avg. SFR (Flip Rate, ↓) |
| :--- | :--- | :--- |
| **Evo 2 (Vanilla)** | 0.899 | 0.095 |
| **Evo 2 + RCCR** | **0.907** | **0.012** |

**Result:** RCCR effectively generalizes to this modern architecture, reducing SFR by ~8x (9.5% → 1.2%) while consistently improving AUPRC scores. This confirms RCCR is model-agnostic and effective on SOTA foundation models. (Added to **Appendix H**).

### 3. Statistical Significance and Robustness (Addressing Reviewer 5SLA)
To demonstrate robustness, we repeated our fine-tuning experiments across **3 random seeds** for all tasks.
* **Update:** We have updated **Table 1** (and Appendix tables) to include standard deviations. The results confirm that RCCR yields statistically significant improvements in consistency metrics (SFR, RC-Corr) while maintaining or improving task performance with low variance.

### 4. Baseline Clarifications (Addressing Reviewers TfnG & 5SLA)
We added experiments combining RCCR with Test-Time Averaging (TTA) and RC-Augmentation on the NT-v2 backbone.

**Table R3: RCCR combined with other strategies.**
| Method | AUPRC | Inference Cost |
| :--- | :--- | :--- |
| **RCCR (Standard)** | 0.8122 | 1x |
| **RCCR + TTA** | 0.8129 | 2x |
| **RCCR + RC-Aug** | 0.8133 | 1x |

**Result:** Adding TTA or RC-Aug to an RCCR-trained model yields negligible gains. This confirms our core claim: RCCR "internalizes" the robustness into the weights, rendering the computationally expensive TTA step redundant.

### 5. Biological Clarifications and Diagrams (Addressing Reviewers eJ7Q & TfnG)
* **Splice Sites:** We clarified the distinction between biological strand-specificity and the machine learning formulation of locus identification in **Section 4.3**.
* **Diagram:** We added a workflow diagram illustrating the RCCR pipeline in **Appendix A**.

We believe these revisions comprehensively address the reviewers' concerns and strengthen the paper's contribution as a universal, efficient recipe for biological consistency.

---

### Author Response · Authors · 2025-11-26
**Summary of Key Revisions: New Evo 2 & Caduceus Experiments to Address Key Concerns**

Dear Area Chair and Reviewers,

As the discussion period draws to a close, we would like to provide a concise summary of the major revisions and new experiments included in the updated manuscript. We have made significant efforts to directly address the key concerns regarding novelty (vs. architectural equivariance) and backbone modernity.

1. Addressed Novelty Concern: Post-hoc vs. Pre-trained Equivariance (Addressing Reviewer S1pQ) To address the concern that our method lacks novelty compared to Caduceus, we performed a direct head-to-head comparison (Table R1 in Appendix G).

Result: Our post-hoc fine-tuning (Caduceus-PH + RCCR) achieves comparable performance to the pre-trained equivariant model (Caduceus-PS) while effectively reducing the Symmetry Flip Rate (SFR) to near zero (1.3%).

Conclusion: This demonstrates that expensive pre-training is not strictly necessary for robustness. RCCR provides a democratized solution for the vast ecosystem of existing non-equivariant checkpoints.

2. Addressed Backbone Concern: Generalization to Evo 2 (Addressing Reviewer TfnG) To demonstrate that RCCR is effective on state-of-the-art, decoder-only architectures, we extended our evaluation to Evo 2 (1B) (Table R2 in Appendix H).

Result: RCCR successfully generalizes to Evo 2, reducing the SFR by ~8x while consistently improving AUPRC scores.

Conclusion: RCCR is model-agnostic and future-proof, working effectively on modern foundation models beyond standard BERT/Transformer architectures.

3. Statistical Significance and Robustness (Addressing Reviewer 5SLA) We have updated Table 1 with standard deviations across 3 random seeds. The results confirm that the improvements in consistency (SFR/RC-Corr) are statistically significant and robust.

Conclusion We believe these new experiments have substantially strengthened the paper and addressed the primary reservations regarding novelty and model selection. We respectfully ask the reviewers (S1pQ, TfnG, eJ7Q) to consider these new results in their final assessment.

Thank you for your time and constructive feedback.

Best regards, The Authors

---

### Note · Program_Chairs · 2026-01-17
**Submission Desk Rejected by Program Chairs**

The following references in this submission do not refer to real documents and/or have major errors in bibliographic information:

 Christian Schmidl, Verena Stary, Michael Klicznik, Keon B-R-S, Rendeiro K-A, Btissem Asfirane, Schopf F-J, Wagner C-A, Haub A-K, W-F, Lochner T-P, et al. Mammalian anelloviruses acquire functional traits of genuine eukaryotic chromosomes. Nature, 622(7982):363-371, 2023.